**RESEARCH**                                                                                    **Open Access**

# An interpretable bimodal neural network characterizes the sequence and preexisting chromatin predictors of induced transcription factor binding

Divyanshi Srivastava[1], Begüm Aydin[2], Esteban O. Mazzoni[2] and Shaun Mahony[1*]

* Correspondence: mahony@psu.edu
[1]Center for Eukaryotic Gene Regulation, Department of Biochemistry & Molecular Biology, Pennsylvania State University, University Park, PA, USA
Full list of author information is available at the end of the article

## Abstract

**Background:** Transcription factor (TF) binding specificity is determined via a complex interplay between the transcription factor's DNA binding preference and cell type-specific chromatin environments. The chromatin features that correlate with transcription factor binding in a given cell type have been well characterized. For instance, the binding sites for a majority of transcription factors display concurrent chromatin accessibility. However, concurrent chromatin features reflect the binding activities of the transcription factor itself and thus provide limited insight into how genome-wide TF-DNA binding patterns became established in the first place. To understand the determinants of transcription factor binding specificity, we therefore need to examine how newly activated transcription factors interact with sequence and preexisting chromatin landscapes.

**Results:** Here, we investigate the sequence and preexisting chromatin predictors of TF-DNA binding by examining the genome-wide occupancy of transcription factors that have been induced in well-characterized chromatin environments. We develop Bichrom, a bimodal neural network that jointly models sequence and preexisting chromatin data to interpret the genome-wide binding patterns of induced transcription factors. We find that the preexisting chromatin landscape is a differential global predictor of TF-DNA binding; incorporating preexisting chromatin features improves our ability to explain the binding specificity of some transcription factors substantially, but not others. Furthermore, by analyzing site-level predictors, we show that transcription factor binding in previously inaccessible chromatin tends to correspond to the presence of more favorable cognate DNA sequences.

**Conclusions:** Bichrom thus provides a framework for modeling, interpreting, and visualizing the joint sequence and chromatin landscapes that determine TF-DNA binding dynamics.

## Background

Sequence-specific transcription factors (TFs) bind DNA at their cognate sequence motifs, using both direct base interactions and DNA structural feature recognition [1–3]. However, the presence of cognate motif instances alone is a poor predictor of TF binding [4, 5]. TFs typically bind a small fraction of their potential target motif instances in a given cell type, and the cohort of sites which are bound can vary greatly across cell types [6–8]. These observations suggest that cell type-specific TF selectivity is governed by cell type-specific chromatin environments [5, 7, 9, 10]. Cell-specific patterns of chromatin accessibility, nucleosome positioning, and histone post-translational modifications modify the availability of a TF's sequence motifs [11–16]. Co-operative interactions with other regulatory proteins can alter a TF's intrinsic sequence preferences, or enable binding at otherwise unavailable target sequence motifs [5, 17–19]. Even pioneer TFs, which are characterized by their ability to bind target motifs in relatively inaccessible chromatin, bind DNA in cell type-specific patterns that can be modulated by other TFs [20–23]. Thus, it remains unclear how DNA sequence, chromatin structure, and interactions with other regulators act in concert to determine cell type-specific TF binding patterns.

Most previous searches for features associated with cell type-specific TF binding sites have performed correlations with chromatin data measured when the TFs under study are already bound to DNA (i.e., "concurrent" chromatin information) [6, 24–30]. But TFs and their recruited regulatory complexes often alter local chromatin landscapes [31, 32]. Concurrent chromatin features thus cannot be used to address the question of how TF binding patterns become established in the first place. The few studies that have analyzed the determinants of TF binding in dynamic contexts have lacked integrated analysis approaches that separate DNA sequence and prior chromatin predictors of future TF binding activities [9, 11, 12, 21, 33, 34].

We present Bichrom, a bimodal neural network framework for characterizing the relative contributions of DNA sequence and preexisting cell type-specific chromatin landscape to an induced TF's binding specificity (Fig. 1a). Our use of neural networks is motivated by their advantages in predicting genome-wide TF binding patterns [24, 28, 35], and the ability of multi-modal neural networks to integrate heterogeneous data types [36–38]. Bichrom's architecture embeds TF binding sites into a two-dimensional latent space, which can be used to estimate the relative contributions of DNA sequence and preexisting chromatin features at individual TF binding sites. By comparing how well neural networks can represent genome-wide binding patterns using sequence information alone versus a combination of sequence and preexisting chromatin features, we can quantify the marginal amount of information added by preexisting chromatin. Comparing such metrics across TFs allows us to assess how TFs differ in their overall sensitivity to preexisting chromatin.

Our approach is distinct from other recent applications of neural networks to TF binding prediction tasks. Several studies have also used neural networks to integrate DNA sequence and concurrent chromatin landscape information, but with the goal of imputing unobserved TF binding patterns in a given cell type [24, 28, 39]. In contrast, we focus on settings in which we already know (via ChIP-seq) where a TF of interest is binding when expressed in a given cell type, and we aim to interpret how those binding sites relate to DNA sequence and preexisting chromatin features. Due to their focus on

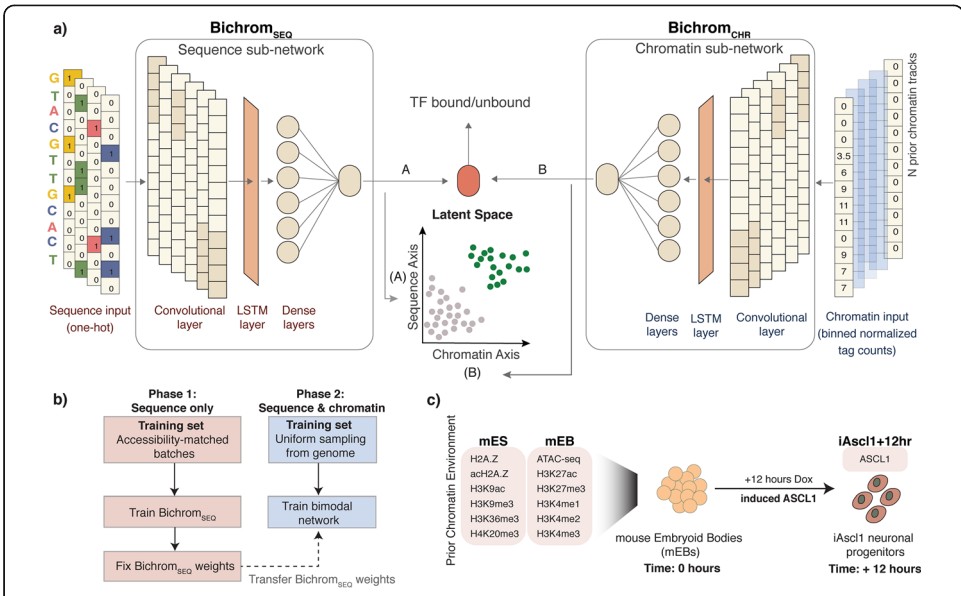

**Fig. 1** Overview of Bichrom's neural network architecture and approach. **a** Bichrom's bimodal sequence and preexisting chromatin network consists of two sub-networks: the sequence sub-network (Bichrom$_{SEQ}$) which uses one-hot encoded DNA sequence as input; and the chromatin sub-network (Bichrom$_{CHR}$) which uses binned normalized tag counts from chromatin experiments such as ATAC-seq and histone modification ChIP-seq. The sequence and chromatin sub-network activations embed the training data into a lower-dimensional plane, which is then used by a sigmoid-activated node for TF binding label classification (i.e., bound/unbound). **b** Overview of Bichrom's training strategy. Bichrom$_{SEQ}$ is trained using training batches within which positive and negative training samples are matched in their prior accessibility status. The weights of the convolutional and LSTM layers of Bichrom$_{SEQ}$ are fixed, and Bichrom is trained using both sequence and preexisting chromatin data. **c** Overview of the Ascl1 data: Ascl1 expression is induced in mouse embryoid bodies (mEBs) using a Dox-inducible promoter and Ascl1 binding is measured 12 h post induction. Bichrom training uses 12 prior chromatin datasets from mEB and mES cell types

imputation or prediction tasks as opposed to feature attribution, previous neural network applications use early-integration frameworks (i.e., DNA sequence and chromatin accessibility data are integrated at a feature level) [6, 24, 26, 39–41]. However, early-integration makes it challenging to interpret the contributions of individual features toward TF binding at individual sites. Our bimodal architecture has the distinct advantage of enabling deconvolution of sequence and chromatin predictors of TF-DNA binding.

We demonstrate Bichrom's utility by examining the binding determinants of the proneural bHLH TF Ascl1 when it is induced in mouse embryoid body (mEB) cells [42]. Bichrom analysis finds that despite Ascl1's characterized pioneering abilities [42–46], Ascl1 binding is dependent on the preexisting mEB chromatin environment at a subset of its binding sites. Next, we demonstrate that Bichrom can informatively rank TFs by their relative dependence on prior chromatin landscapes. We focus on a selection of neuronal TFs that are activated downstream of Ascl1 (and bind DNA in the chromatin environment established by Ascl1) [42]; Bichrom's assessment of relative chromatin dependence for these TFs is supported by observing whether the same TFs bind different sites when they become activated in a different chromatin environment. Finally, we expand our analysis to examine the differential sequence and prior chromatin drivers for 12 TFs induced in cell types for which the preexisting chromatin accessibility landscape

has also been characterized (mouse and human fibroblasts [23, 47]). While we focus here on systems in which TF expression is induced in cell lines, Bichrom is broadly applicable to study TF binding determinants in any dynamic regulatory system in which chromatin landscapes can be assayed before a TF's DNA binding activity occurs.

## Results

### A bimodal neural network integrates DNA sequence and preexisting chromatin data to characterize TF binding predictors

Bichrom's bimodal neural network is composed of two sub-networks: one that operates only on DNA sequence features (Bichrom$_{SEQ}$), and one that operates on chromatin features derived from ATAC-seq and ChIP-seq specific for histone modifications (Bichrom$_{CHR}$). Both sub-networks are designed to output real-valued activations, which are additively combined by a single sigmoid-activated node (Fig. 1a). Bichrom is trained to predict TF binding labels, as defined by ChIP-seq peaks. In our typical usage, Bichrom trains on binding labels for a TF that has become activated in a given cell type, using input features from DNA sequence and chromatin data profiled before the targeted TF has become active. While we phrase Bichrom training as a predictive task, our motivation is not to predict TF binding (which, of course, has already been observed via ChIP-seq), but rather to characterize the sequence and preexisting chromatin states that define a TF's binding pattern. We do so by taking advantage of the interpretable nature of the bimodal network architecture; the weighted activations flowing from each sub-network result in a latent two-dimensional representation of each genomic region, which we can analyze to interpret and compare the sequence and preexisting chromatin predictors at individual TF binding sites (Fig. 1a).

Bichrom's neural network hyper-parameters were chosen via a random grid-search (see "Methods"). Each sub-network consists of a single convolutional neural network (CNN) layer that acts as a primary feature extractor, followed by a long short-term memory (LSTM) layer that can capture potential interactions between convolutional filters [24, 48]. The LSTM output is fed through dense layers, which are combined to produce a scalar real-valued output (Fig. 1a). Bichrom's CNN-LSTM architecture was found to perform better than or equivalent to a wide range of alternative CNN-based architectures (see "Methods," Additional file 1: Fig. S1). Each sub-network is input features from 500 bp genomic windows: Bichrom$_{SEQ}$ operates on one-hot encoded DNA sequences, while Bichrom$_{CHR}$ operates on normalized and binned read counts from one or more pre-existing chromatin data tracks. Since we aim to interpret the sub-network activations as separable sources of sequence and prior chromatin information, we wish to avoid scenarios where the Bichrom$_{SEQ}$ sub-network learns sequence features that are associated with the prior chromatin landscape. To minimize such confounding issues, we train Bichrom$_{SEQ}$ using mini-batches where the positive and negative sets are matched in their prior accessibility labels (see "Methods," Fig. 1b).

While our primary motivation is the interpretation of sequence and prior chromatin predictors of induced TF binding, we asked whether Bichrom's architecture and training scheme produces an accurate representation of genome-wide TF-DNA binding data. To ensure that Bichrom can represent genome-wide TF binding patterns with accuracies approaching current state-of-the-art predictive methods, we assessed

Bichrom's performance on within-cell-type predictive tasks from the ENCODE-DREAM challenge [13, 24, 25, 28]. This challenge evaluated the ability of various methods to predict genome-wide binding of several human TFs given concurrent chromatin accessibility (DNase-seq) and gene expression data. Using only sequence and concurrent chromatin accessibility data, Bichrom predicts TF-DNA binding in held-out genomic regions with accuracies comparable to, albeit slightly lower than, the top models described in the challenge (see "Methods," Additional file 1: Fig. S2). Thus, Bichrom's bimodal architecture does not negatively impact its ability to represent genome-wide TF-DNA binding patterns using sequence and chromatin features.

### Ascl1 binding sites are partially predicted by preexisting chromatin information

Ascl1 is a proposed pioneer TF which can bind to relatively inaccessible sites when induced in fibroblasts and early embryonic cell types [42–46]. Whether the preexisting chromatin environment plays a role in determining induced Ascl1 binding sites remains unclear [44]. To address this question, we applied Bichrom to interpret the relationships between induced Ascl1 binding and preexisting sequence and chromatin landscapes. Specifically, we train Bichrom using previously published Ascl1 ChIP-seq data measured 12 h after Ascl1 expression has been induced in mouse embryoid body (mEB) cells [42] (Fig. 1c). The 12-h timepoint is the earliest at which we can obtain robust Ascl1 ChIP-seq binding data and is thus the most likely to represent the initial binding activities that are shaped by previous chromatin states. In this analysis, the $Bichrom_{CHR}$ sub-network is trained using 12 chromatin-related datasets from mEB and related mouse embryonic stem (mES) cells: ATAC-seq, H2A.Z, acH2A.Z, H3K27ac, H3K27me3, H3K4me1/me2/me3, H3K9ac, H3K9me3, H3K36me3, and H4K20me3 (Additional file 2: Table S1, Additional file 2: Table S2, Fig. 1c).

To assess whether the preexisting chromatin landscape is predictive of future Ascl1 binding locations, we compared Bichrom performance with a baseline neural network trained only with sequence information. The baseline sequence-only network was constructed using the same hyper-parameters and architecture as the $Bichrom_{SEQ}$ sub-network. Training was repeated 9 times for each network, each training round using a separate held-out test chromosome. The sequence-only network predicts induced Ascl1 binding with a median area under the precision-recall curve (auPRC) of 0.42. In contrast, Bichrom predicts induced Ascl1 binding with a median auPRC of 0.59, suggesting that information in the preexisting chromatin landscape significantly improves prediction of Ascl1 binding (Wilcoxon signed rank test $p$ value 0.003, Fig. 2a). As a negative control, Bichrom trained using sequence and a ChIP input control experiment instead of preexisting chromatin data does not lead to significant improvement in network performance when compared to a sequence-only network (auPRC = 0.45, Fig. 2a). Additionally, we confirmed that Bichrom's additive bimodal design does not perform worse than a model with more complex interactions between sequence and prior chromatin features (Additional file 1: Fig. S3B).

Notably, the improved performance of Bichrom's bimodal network is driven largely by improved specificity. At a false positive rate of 0.05, a majority of Ascl1-bound sites are correctly predicted by both the sequence-only network and Bichrom (Additional file 1: Fig. S3A). However, at a fixed recall of 0.5, Bichrom's precision is substantially

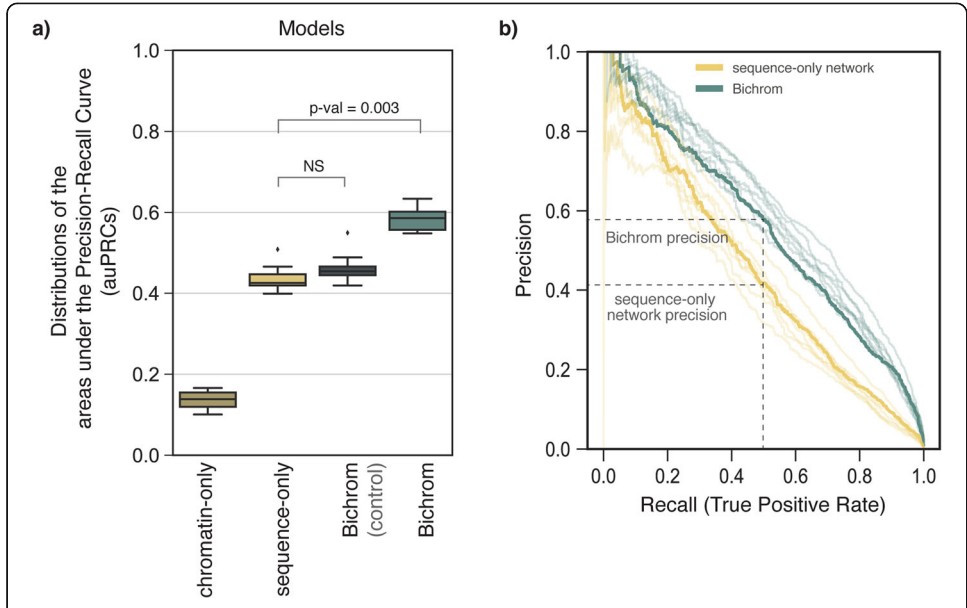

**Fig. 2** Preexisting chromatin information improves Bichrom predictions of induced Ascl1 binding sites. **a** Distribution of model performance (auPRC) in predicting Ascl1 binding sites for a neural network trained using only chromatin data (chromatin-only), a neural network trained using only sequence data (sequence-only), Bichrom using sequence and a ChIP input control experiment (Bichrom control), and Bichrom using sequence and 12 preexisting ES chromatin datasets. The boxplots represent data from 9 independent training sets, each consisting of a separate held-out test chromosome. **b** The precision-recall curves for 9 models, each tested on 9 distinct held-out test chromosomes. The P-R curves for networks that use a training set in which chromosome 10 is held-out for testing are highlighted in solid lines, performance for other training sets is represented with lighter (alpha = 0.2) traces. Precision at a fixed recall of 0.5 is highlighted for both the sequence-only network and Bichrom

greater than that of the sequence-only network (Fig. 2b). Thus, the incorporation of mEB and mES chromatin information significantly improves Bichrom's ability to predict induced Ascl1 binding in a future timepoint, suggesting that although it is an established pioneer TF, Ascl1 binding sites are partially determined by the preexisting chromatin landscape.

## Bichrom deconvolves the sequence and preexisting chromatin predictors of induced Ascl1 binding

Beyond quantifying preexisting chromatin's overall contribution to improving Ascl1 binding predictions, Bichrom's unique bimodal architecture enables decomposition of sequence and prior chromatin predictors at individual Ascl1 binding sites. For a given genomic window, Bichrom's TF binding score is a simple linear combination of the activations from $Bichrom_{SEQ}$ and $Bichrom_{CHR}$ sub-networks. Thus, every genomic window can be embedded in a two-dimensional latent space defined by the sub-network activations, enabling an intuitive visualization of how much the sequence and prior chromatin sub-networks contributed to the overall predictive score (Fig. 1a).

Applied to the Bichrom network trained on Ascl1 binding data, we find that Ascl1-bound genomic windows (orange) are well-separated from randomly sampled unbound genomic windows (gray) in the two-dimensional latent space (Fig. 3a). However, Ascl1 binding sites are distributed over a broad range of $Bichrom_{SEQ}$ and $Bichrom_{CHR}$ sub-

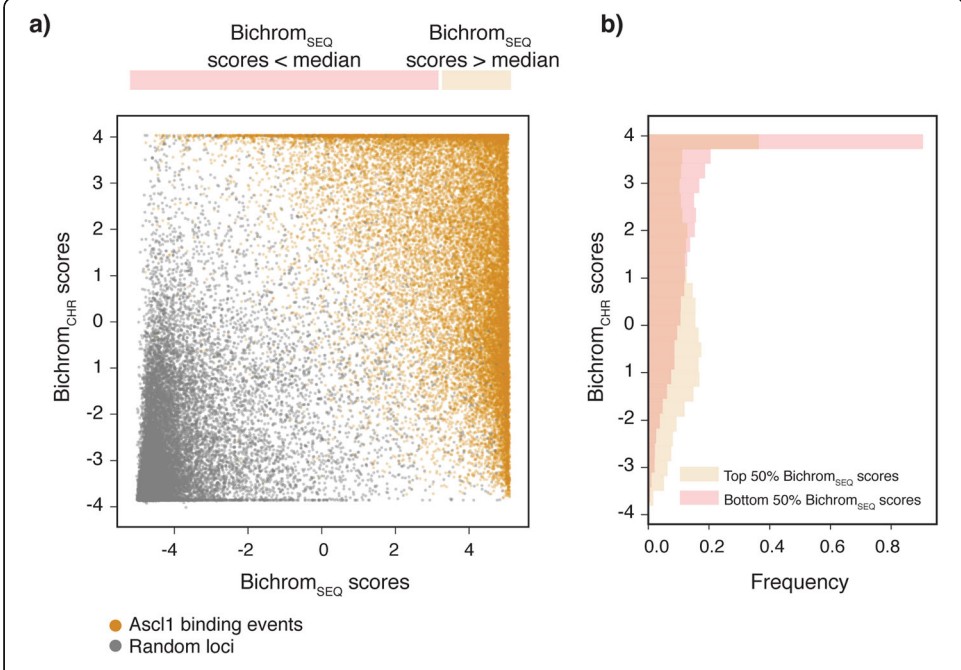

**Fig. 3** Bichrom decomposes sequence and preexisting chromatin predictors at individual Ascl1 binding sites. **a** Bichrom network-derived latent representation of induced Ascl1 binding events (orange) and a randomly sampled subset of unbound genomic regions (gray). The axes represent contributions of the Bichrom$_{SEQ}$ sub-network and the Bichrom$_{CHR}$ sub-network toward bimodal network predictions. **b** Distributions of Bichrom$_{CHR}$ scores for Ascl1 binding events conditioned on the Bichrom$_{SEQ}$ scores

network activation scores. The variation in sub-network activations suggests that some of Bichrom's Ascl1 binding site predictions are driven primarily by sequence information, while some are driven by preexisting chromatin features (Fig. 3a). Furthermore, the diversity in Ascl1 sequence sub-network (Bichrom$_{SEQ}$) scores is higher at sites that are scored favorably by the Bichrom$_{CHR}$ chromatin sub-network, suggesting that the presence of a favorable preexisting chromatin environment may enable Ascl1 binding even at sites with relatively weaker sequence features (Fig. 3b). We confirm that this compensatory effect is not an artifact of the linear combination between the sequence and chromatin sub-networks by fixing the sequence sub-network weights while training the chromatin sub-network (see "Methods," Additional file 1: Fig. S5). Thus, Bichrom learns a model in which Ascl1 binding sites exhibit a broad range of sequence and chromatin sub-network scores, where favorable prior chromatin features can partially compensate for weaker sequence features.

To investigate informative sequence features at Ascl1 binding sites, we used an integrated gradients [49] feature attribution-based approach to identify local sequence windows driving high Bichrom$_{SEQ}$ sub-network scores (see "Methods," Additional file 1: Fig. S6A, B). As expected, regions with high Bichrom$_{SEQ}$ scores contain motifs related to the CAGSTG E-box, consistent with Ascl1's cognate DNA binding preference [42] (Fig. 4a). We also find a POU homeodomain DNA binding motif, suggesting that Ascl1 may bind a subset of its sites either in concert with, or at sites pre-bound by, a POU domain TF such as Oct4 (one of the main regulators of pluripotency in the preexisting

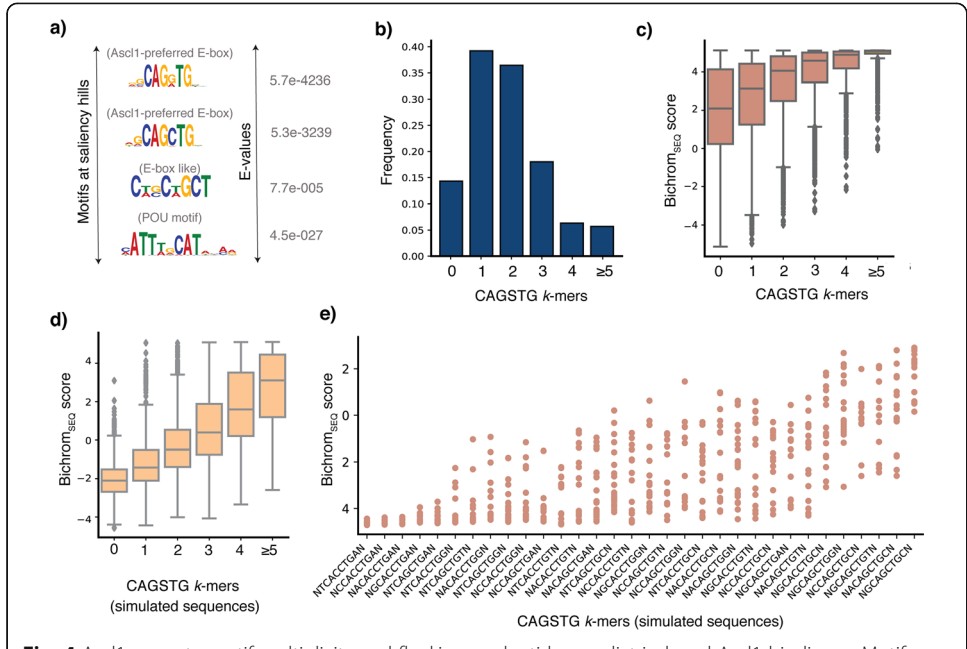

**Fig. 4** Ascl1 cognate motif multiplicity and flanking nucleotides predict induced Ascl1 binding. **a** Motifs enriched at saliency hills (regions of the input sequences that are used by the sequence sub-network to make predictions). **b** Frequencies of CAGSTG *k*-mers at all Ascl1 binding sites. **c** Bichrom$_{SEQ}$ scores increase with increasing motif multiplicity at Ascl1 binding windows. **d** Embedding CAGSTG motifs in simulated sequences confirms that the Bichrom$_{SEQ}$ sub-network uses the number of motif occurrences as a score-driving predictor. **e** Bichrom$_{SEQ}$ scores assigned to CAGSTG *k*-mers vary widely based on nucleotides flanking the cognate *k*-mer

embryonic cells) (Fig. 4a). Bichrom$_{SEQ}$ scores are strongly associated with Ascl1 cognate motif multiplicity; Ascl1 binding sites that receive high Bichrom$_{SEQ}$ scores contain higher frequencies of E-box motif instances (Additional file 1: Fig. S6C) and CAGSTG *k*-mers (Fig. 4b, c). The correlation between motif multiplicity and Bichrom$_{SEQ}$ scores is maintained when scoring randomly sampled sequences in which we inserted variable numbers of CAGSTG *k*-mer instances (Fig. 4d). We also investigated whether Bichrom$_{SEQ}$ scores encapsulate subtle nucleotide composition dependencies in regions flanking Ascl1's cognate binding motif [2, 42, 50]. Specifically, we inserted CAGSTG *k*-mers flanked by variable nucleotides into an artificial uniform sequence (see "Methods") and found that Bichrom$_{SEQ}$ scores vary substantially according to specific motif-flanking nucleotide compositions (Fig. 4e).

We next investigated the preexisting chromatin features driving variability in Bichrom$_{CHR}$ sub-network scores. Ascl1 binding sites that receive the highest Bichrom$_{CHR}$ scores are enriched in chromatin signals associated with regulatory activity in the preexisting mES and mEB cell types (e.g., preexisting ATAC-seq, H3K27ac, H3K4me2, H3K4me3, H3K9ac; Fig. 5a). Further, genome-wide domains of preexisting ATAC-seq and "active" histone mark ChIP-seq enrichment are more likely to receive higher Bichrom$_{CHR}$ scores (Fig. 5b). H3K9me3 and H4K20me3 domains receive the lowest Bichrom$_{CHR}$ scores, in contrast to a previous suggestion that preexisting H3K9me3 is predictive of induced Ascl1 binding in mouse embryonic fibroblasts [44] (Fig. 5b).

Consistent with our analyses of individual chromatin signals, Ascl1 binding sites that overlap various ChromHMM-defined [51] mES/mEB chromatin states receive widely varying median Bichrom$_{CHR}$ scores (Fig. 5c). Ascl1 binding sites that overlap preexisting active promoter and strong enhancer states receive the highest Bichrom$_{CHR}$ scores, while Polycomb-repressed (marked by H3K27me3), heterochromatin (marked by H3K9me3), and quiescent states receive the lowest Bichrom$_{CHR}$ scores (Fig. 5c). In accordance with our observations from the latent space embeddings (Fig. 3), Ascl1 binding sites located in preexisting enhancer or promoter states receive lower median scores from the Bichrom$_{SEQ}$ sequence sub-network (Fig. 5c). Conversely, Ascl1 sites in preexisting heterochromatin and quiescent states receive the highest median Bichrom$_{SEQ}$ scores.

Taken together, our Bichrom analysis of induced Ascl1 binding sites suggest mutual compensation between sequence and preexisting chromatin predictors. Ascl1 binding to sites of preexisting active chromatin does not necessarily require strong cognate and secondary sequence motif features, whereas Ascl1 binding to preexisting quiescent or repressed chromatin is correlated with increased motif multiplicity and favorable flanking nucleotide composition.

### Bichrom predicts the relative dependence of neuronal TF binding sites on preexisting chromatin

As demonstrated by our applications to Ascl1 data, Bichrom's analyses can be interpreted as characterizing a TF's dependence on preexisting chromatin, both from a global viewpoint and at the level of individual binding sites. We next asked whether we could support such interpretations experimentally. To do so, we apply Bichrom to analyze the binding patterns of TFs that become expressed downstream of Ascl1 in the same mEB-based system [42].

Specifically, induced Ascl1 (iAscl1) differentiates mEBs into neuronal lineages [42]. Within 12 h of expression, Ascl1 binding establishes chromatin accessibility at some previously inaccessible binding sites. Ascl1 induces expression of several key neuronal TFs, including Brn2, Ebf2, and Onecut2, within this new chromatin accessibility landscape. Having previously characterized the subsequent genomic binding of Brn2, Ebf2, and Onecut2 using ChIP-seq at iAscl1 + 48 h [42], we applied Bichrom to ask whether each TF's binding is informed by the chromatin accessibility landscape established by iAscl1 (ATAC-seq, iAscl1 + 12 h).

Compared with a network trained using sequence information alone, Bichrom's incorporation of preexisting iAscl1 + 12 h accessibility data significantly improves the representation of iAscl1 + 48 h binding sites for both Brn2 and Ebf2. Specifically, we observe improvements in recall at a fixed false positive rate (see "Methods," Fig. 6a) and auPRC (Brn2: sequence network auPRC = 0.23, Bichrom auPRC = 0.35; Ebf2: sequence network auPRC = 0.38, Bichrom auPRC = 0.53; Additional file 1: Fig. S7). In contrast, the incorporation of prior chromatin information does not result in improved representation of Onecut2 binding. The sequence-only network and Bichrom show comparable values for both recall (Fig. 6a) and auPRC (sequence network auPRC = 0.54, Bichrom auPRC = 0.56, *p* value 0.59) (Additional file 1: Fig. S7). Therefore, Bichrom analysis suggests that Ebf2's and Brn2's genomic binding specificity is

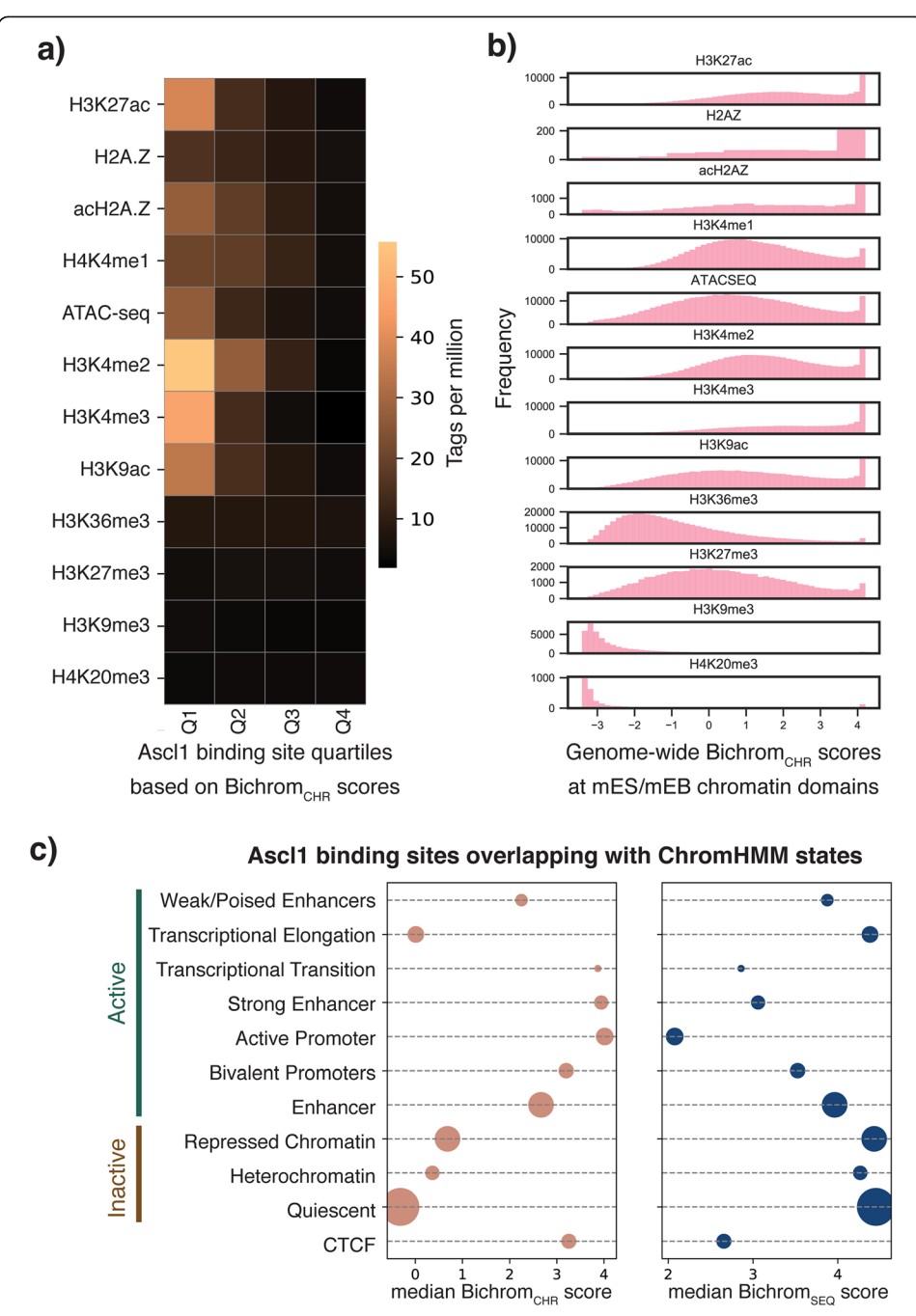

**Fig. 5** Preexisting active chromatin features positively predict induced Ascl1 binding sites. **a** Mean mES/mEB chromatin feature tag enrichment at Ascl1 binding sites, divided into quartiles based on their association with Bichrom$_{CHR}$ sub-network scores. Sites belonging to the highest-scoring quartile are enriched for preexisting active histone modifications such as H3K4me2, H3K4me3, H3K27ac, and H3K9ac. The lowest-scoring quartile lack enrichment of any measured histone modifications. **b** The distribution of Bichrom$_{CHR}$ sub-network scores at genomic regions enriched for each of the mES/mEB chromatin features. **c** Median Bichrom$_{SEQ}$ and Bichrom$_{CHR}$ sub-network scores at Ascl1 sites that overlap mES/mEB ChromHMM states. Bubble size corresponds to relative proportion of Ascl1 binding sites. Median Bichrom$_{SEQ}$ sub-network scores are typically lower at states with high Bichrom$_{CHR}$ sub-network scores

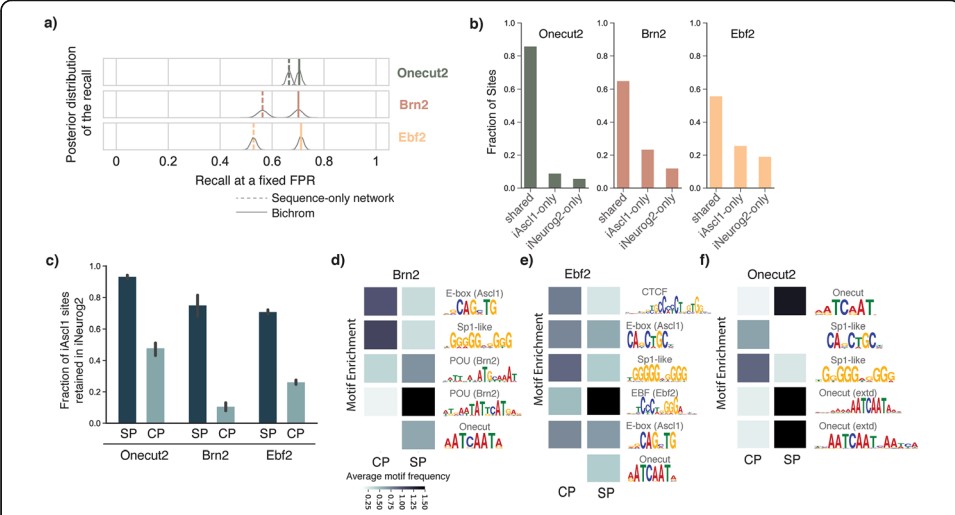

**Fig. 6** Brn2, Ebf2, and Onecut2 vary in their relative dependence on preexisting chromatin features. **a** Bichrom (incorporating preexisting ATAC-seq data) outperforms the sequence-only network at predicting induced Brn2 and Ebf2 binding but performs at par with the sequence-only network at predicting induced Onecut2 binding. **b** The percentage of Onecut2, Brn2, and Ebf2 sites that are bound in both iAscl1 and iNeurog2 neurons (shared sites), preferentially bound in iNeurog2 (iNeurog2-only) and preferentially bound in iAscl1 neurons (iAscl1-only). **c** Brn2, Ebf2 and Onecut2 iAscl1 sequence-predicted (SP) sites are more conserved in iNeurog2 neurons compared to chromatin-predicted (CP) sites. MEME-ChIP-determined motif frequency at SP vs. CP sites for **d** Brn2, **e** Ebf2, and **f** Onecut2. For all three TFs, SP sites contain a higher number of cognate motifs. CP and SP sites also display differential secondary motif enrichment

dependent on preexisting accessibility to a greater degree than that of Onecut2. In other words, Bichrom predicts that Onecut2 has a more pioneer-like behavior than Ebf2 and Brn2.

If Onecut2 is in fact less dependent on preexisting chromatin accessibility, we should expect it to bind many of the same sites in a different chromatin accessibility landscape. Similar to Ascl1, expression of the proneural bHLH TF Neurog2 in mEB cells also leads to differentiation along neuronal lineages and also causes expression of Brn2, Ebf2, and Onecut2 [42]. However, induced Neurog2 (iNeurog2) binds to different sites and thereby establishes a distinct chromatin accessibility landscape (Additional file 1: Fig. S8). Consistent with our interpretation of Bichrom's analysis, the iNeurog2 + 48 h binding activities of Brn2 and Ebf2 are affected by the shift in preexisting chromatin accessibility landscapes to a greater degree than Onecut2. Specifically, only 65% of Brn2 sites and 56% of Ebf2 sites are shared between iAscl1 and iNeurog2 cell types, whereas 86% of Onecut2 binding sites are shared (Fig. 6b) [42]. This comparative analysis of differential binding across two cell types thus supports Bichrom's ranking of the TFs' dependence on preexisting chromatin, which was derived from analysis of ChIP-seq data from a single cell type (i.e., iAscl1 + 48 h). Furthermore, Bichrom's prediction that Onecut2 is less dependent on preexisting chromatin is consistent with recent literature that establishes Onecut TFs as neuronal pioneers [52, 53].

Turning to the predictors of individual TF binding sites, Bichrom's two-dimensional latent space embeddings again show wide ranges of Bichrom$_{SEQ}$ and Bichrom$_{CHR}$ sub-network scores for all three neuronal TFs (Additional file 1: Fig. S9). For each TF, we extract subsets of binding sites that we define as "sequence-predicted" (SP; sites in top

25th percentile of Bichrom$_{SEQ}$ scores and bottom 25th percentile of Bichrom$_{CHR}$ scores) and "chromatin-predicted" (CP; sites in top 25th percentile of Bichrom$_{CHR}$ scores and bottom 25th percentile of Bichrom$_{SEQ}$ scores). If our latent space embeddings are meaningful, we would expect the "CP" sites to be more dependent on a shift in the preexisting chromatin landscape than the "SP" sites. Analysis of differential binding downstream of iNeurog2 again supports our interpretation. For all three neuronal TFs, "SP" iAscl1 binding sites show consistently higher levels of retention in iNeurog2 cells than "CP" sites (Fig. 6c). For example, 74% of Brn2's iAscl1 SP sites are also bound in iNeurog2 cells, while only 11% of Brn2's iAscl1 CP sites are retained (Fig. 6c). Thus, Bichrom's prediction that some individual binding sites are more dependent on preexisting chromatin features is confirmed by those sites being more sensitive to a shift in the underlying chromatin landscape.

Our interpretations of the latent embeddings are also supported by motif analysis at SP and CP sites (Fig. 6d–f). The cognate DNA-binding motifs for Brn2, Ebf2, and Onecut2 show consistently higher enrichment in the SP sites compared with their CP sites (Fig. 6d–f). In contrast, and even though the Bichrom$_{CHR}$ sub-network does not use any DNA sequence information, we find that all three TF's CP sites contain higher enrichment for a Sp1-like motif and the CAGSTG E-box motif that is preferred by Ascl1. The latter observation suggests that some CP sites bound by the neuronal TFs at 48 h may be made accessible by Ascl1 binding at 12 h. Indeed, we find that CP sites overlap preexisting Ascl1 sites (iAscl1 + 12 h) at a significantly higher rate than SP sites: for example, 59% of Brn2's CP sites overlap preexisting Ascl1 binding compared to 1% of Brn2 SP sites. Similarly, we find a CTCF-like motif enriched at higher rates in Ebf2 CP sites than SP sites (Fig. 6e), suggesting that CTCF binding in a prior cell stage may establish a favorable chromatin environment for Ebf2 binding. Comparing with mES CTCF ChIP-seq data, we find that 33% of Ebf2 CP sites overlapped with CTCF binding events, while fewer than 1% of Ebf2 SP sites do so.

In summary, the latent network embeddings identify sites that are more likely to be differentially bound in distinct chromatin environments. Additionally, they can be used to identify the diverse sequence, chromatin, and co-factor feature compositions that specify genome-wide TF binding.

### The predictive capacity of preexisting chromatin varies across TFs

Finally, we asked whether the ability of preexisting chromatin to explain binding specificity varies across a broader range of TFs, cellular conditions, and datasets. We applied Bichrom to analyze the binding of 12 induced TFs from studies where aspects of the preexisting chromatin environment were also characterized. Specifically, we used Bichrom to analyze the binding of 9 TFs that were induced in mouse NIH-3T3 fibroblasts and assayed approximately 12 h post TF induction (Additional file 2: Table S3) [47], using NIH-3T3 ATAC-seq to define the preexisting chromatin accessibility landscape (Additional file 2: Table S4). We also analyzed the binding of three previously established human pioneer TFs—OCT4, GATA4, and FOXA2—that were induced in human BJ fibroblasts and assayed 4 days post induction [23]. In these latter analyses, the preexisting chromatin landscape was defined by ATAC-seq, H3K27ac, H3K4me2, and H3K27me3 (Additional file 2: Table S5).

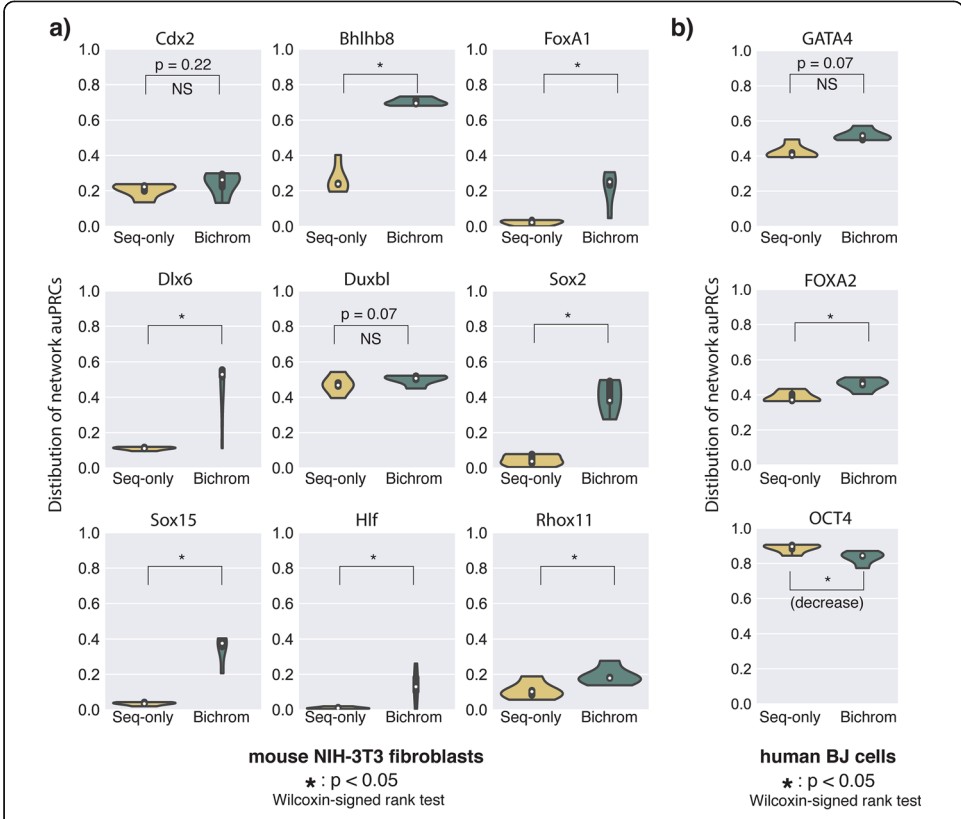

**Fig. 7** Induced TFs display a wide range of relative dependencies on preexisting chromatin features. **a** Distribution of model performance (auPRC) for a sequence-only network and Bichrom (sequence and preexisting NIH-3T3 ATAC-seq) for 9 TFs induced in mouse NIH-3T3 fibroblasts. The boxplots represent data from 5 independent training sets; each consisting of a separate held-out test chromosome. **b** Distribution of model performance (auPRC) for a sequence-only network and Bichrom (sequence and preexisting fibroblast ATAC-seq, H3K27ac, H3K4me2, and H3K27me3) for FOXA2, GATA4, and OCT4 induced in human BJ fibroblasts

Applied to the NIH-3T3-induced TFs, Bichrom's incorporation of preexisting accessibility increased predictive accuracy for NIH-3T3 induced Bhlhb8, Sox15, Sox2, Dlx6, Rhox11, Hlf, and FoxA1 (Fig. 7a, Additional file 1: Fig. S10A). However, the incorporation of preexisting accessibility did not significantly increase predictive accuracy for Duxbl or Cdx2, suggesting that the binding patterns of these TFs are not as dependent on the preexisting chromatin landscape (Fig. 7a, Additional file 1: Fig. S10A). In the human fibroblast datasets, we find the preexisting chromatin data improves Bichrom's predictions of induced TF binding to a limited degree for FOXA1 and GATA4 but does not result in improved predictive capacity for OCT4 (Fig. 7b, Additional file 1: Fig. S10B). Therefore, our computational framework suggests that the preexisting chromatin environment can predict the binding of TFs to varying degrees.

Using Bichrom's latent embeddings of individual binding sites, we interpret the sequence and preexisting chromatin features that predict DNA binding of the examined induced TFs. We find that induced TF binding sites are assigned a broad range of Bichrom$_{SEQ}$ sub-network scores (Additional file 1: Fig. S11). For the majority of TFs, sites scored highly by the Bichrom$_{SEQ}$ subnetwork are associated with increased

cognate motif enrichment (Additional file 1: Fig. S12). For the TFs FoxA1 and Duxbl, higher scoring sites are associated with an increased cognate motif enrichment as well as cognate motifs that contain additional informative flanking nucleotide sequences (Additional file 1: Fig. S12).

Furthermore, for a majority of TFs that bind at least 10% of their sites in previously inaccessible chromatin (Duxbl, Cdx2, Bhlhb8, FOXA2, GATA4), we find that sites bound in pre-inaccessible chromatin are on average assigned higher Bichrom$_{SEQ}$ scores, suggesting that there exists some degree of compensation for these TFs where pre-accessible chromatin enables binding at sites with weaker sequence features (Additional file 1: Fig. S11B). The only TF for which this pattern is reversed is OCT4. Instead of the canonical OCT4 motif, Bichrom$_{SEQ}$ scores are associated with GC-rich sequences (Additional file 1: Fig. S12). In this case, previously accessible sites are assigned higher Bichrom$_{SEQ}$ scores (associated with GC-rich sequences) than previously inaccessible sites, which are enriched for a SOX-related motif (Additional file 1: Fig. S11, Fig. S12). These results are consistent with the observation from Donaghey et al. that 73% of OCT4 ChIP-seq peaks in their human fibroblast inductions overlap CpG islands [23].

Taken together, our analyses of a broad range of induced TFs demonstrate that Bichrom is a useful tool for analyzing the global preexisting chromatin predictors of induced TF binding, and for examining the diversity in sequence and preexisting chromatin features that define individual TF binding sites.

## Discussion

TFs bind subsets of their cognate motif instances in a cell type-specific fashion. Such specificity in TF binding results from an interplay between the TF's inherent sequence preferences and cell type-specific chromatin landscapes [6, 12]. The question naturally arises as to which local chromatin features might enable or inhibit a given TF's DNA binding activities. However, if we can measure a TF's binding occupancy using ChIP-seq, it has by definition already had its own impact on chromatin in that cell type (e.g., by making its binding sites accessible or by recruiting histone modification enzymes). Concurrent chromatin landscapes therefore predict TF binding in the same cell type, but cannot be used to model the causal determinants of that binding.

Bichrom represents an interpretable neural network architecture that can be used to assess the relative contributions of DNA sequence and preexisting chromatin features in specifying an induced TF's genome-wide binding sites. Related to our work, several previous studies have assessed the effects of preexisting chromatin landscapes on the binding of specific TFs [9, 11, 12, 23, 46, 54]. Our work aims to provide a unified predictive framework for quantifying and formalizing the relative contributions of sequence and chromatin predeterminants to TF binding across a range of TFs, both globally and at individual sites. At the global level, we have demonstrated that comparing Bichrom's TF binding predictive performance to that of a sequence-only neural network allows us to infer the degree to which preexisting chromatin shapes a TF's binding landscape. While incorporating prior chromatin data significantly improves predictive performance for some TFs (e.g., Brn2, Dlx6, and Bhlhb8), others seem to display less dependence on prior chromatin (e.g., Onecut2, Duxbl, and Cdx2). Thus, our approach may offer a metric for quantifying the "pioneering" activity of a TF; those that

display less dependence on preexisting chromatin features may have greater "pioneering" abilities.

Although Ascl1 is a well-established pioneer TF [42, 44], Bichrom analysis at the global level suggests that it displays a significant dependence on preexisting chromatin features when induced in mEB cells. In line with this observation, previous work has demonstrated that pioneer TFs bind in context-dependent and cell type-specific patterns [16, 20, 23, 55]. Pioneer TFs may bind nucleosomes preferentially at certain positions with respect to the nucleosome dyad [14, 56], and nucleosomal chromatin may only be bound by pioneer TFs in the presence of distinct favorable motifs [57]. Therefore, it is not entirely surprising that the binding of pioneer TFs like Ascl1 can be shaped by the underlying chromatin landscapes. Our approach aims to quantify such dependencies between global TF binding landscapes and preexisting chromatin environments.

As demonstrated by analysis of Ascl1 binding, the ability of the bimodal network to deconvolve the sequence and prior chromatin features that predict TF binding at individual sites provides a powerful tool to investigate holistic TF binding landscapes. Interpretation of Bichrom-derived latent embeddings at individual binding sites suggests that sequence and preexisting chromatin landscapes are not independent predictors of TF binding. Rather, sequence and preexisting chromatin are mutually compensatory features that define a continuum of sites that may be bound by the induced TF. While genomic loci with weaker sequence signatures may be bound by TFs given a favorable local chromatin environment, the same sequences might not be sufficient to drive TF binding at inaccessible or unfavorable chromatin. For example, Ascl1 is more likely to bind pre-inaccessible loci in the presence of certain sequence features such as high motif multiplicity and favorable motif flanks.

On the other hand, some highly accessible active promoters and enhancers are bound even with weaker sequence signatures, as defined by low activation scores from the $Bichrom_{SEQ}$ sub-network in our model. We note that some TF-bound regions with high preexisting chromatin ($Bichrom_{CHR}$) sub-network activations and low sequence ($Bichrom_{SEQ}$) sub-network activations might represent artifactual ChIP-seq enrichment [58]. Alternatively, these regions may represent direct binding to weaker motifs, or indirect binding mediated by interactions with mES or NIH-3T3 cell regulators [19]. While previous studies have proposed sequence-conditional binding to inaccessible chromatin for a few TFs [12, 41, 57], our work suggests that this compensatory mechanism may exist across a broader range of TFs.

We also allow that our estimates of a TF's dependence on preexisting chromatin may still be cell type-specific as opposed to an innate feature of a given TF. For example, while our analyses suggest that binding sites for the well-characterized pioneer factor FoxA1 are dependent on prior chromatin, this may be specific to the measured context of NIH-3T3 cells. It is possible, for instance, that TFs that cooperate with or otherwise predict FoxA1 binding are already present in NIH-3T3 cells, and FoxA1 may be less dependent on preexisting chromatin in other cell types. We note that for some TFs with only very few identified binding peaks, such as Sox15 and Hlf, the auPRCs for a sequence-only network may be low either due to a highly imbalanced dataset, or due to poorly trained networks. To account for this, we derive posterior distributions for the

network recall. For TFs with fewer peaks calls, these distributions are much wider, reflecting the uncertainty in our estimates.

A few previously described computational methods have aimed to explicitly attribute TF binding predictions to sequence and chromatin features, albeit in the context of concurrent chromatin data [6, 29, 30]. One might expect that such methods could also be applied to interpret relationships between TF binding and preexisting chromatin landscapes. However, Bichrom offers several advantages over previous methods for such tasks. Methods that model concurrent chromatin data often rely on the assumption that chromatin feature distributions are significantly different between TF bound and TF unbound sites. In contrast, preexisting chromatin features are often only weakly associated with induced TF binding, especially for TFs that bind relatively inaccessible chromatin. In Bichrom, we develop training strategies that use step-wise genome-wide training and evaluation strategies to make inferences regarding the predictive capacity of preexisting chromatin on TF binding. Furthermore, some previous approaches for concurrent TF binding analysis used substantially weaker sequence models, featurized by PWM scores or individual *k*-mer counts, as opposed to the more expressive CNN-LSTM model used by Bichrom [29, 30, 35, 48]. The same approaches also mix sequence and chromatin features together in non-additive models such as random forests and support vector machines (SVMs), and hence, the contributions of each feature type (sequence or chromatin) are not directly interpretable at individual sites [29, 30]. In contrast, Bichrom's bimodal neural network architectures are designed to be readily interpretable at individual sites.

Notably, Arvey et al. trained completely independent SVMs on sequence and concurrent chromatin features, and added the outputs to predict TF binding [6]. While such a framework could in principle be applied to score sequence and preexisting chromatin features at individual binding sites, the fact that the sub-models are trained independently makes interpretation of relative contributions challenging. For example, an independently trained sequence SVM could learn sequence features that are generally correlated with active chromatin regions. In contrast, Bichrom's training scheme actively discourages the sequence subnetwork from learning features that are redundant with information provided by preexisting chromatin features.

## Conclusions

Bichrom integrates sequence and preexisting chromatin features using an additive and interpretable neural network framework to predict induced TF binding. Bichrom's additive structure can be used to assess the relative contributions of DNA sequence and preexisting chromatin features in specifying an induced TF's binding sites, both genome-wide and at individual binding sites. We apply Bichrom to several TFs, demonstrating that preexisting chromatin environment is a differential predictor of induced TF binding. Bichrom derived mappings of the predictors of TF binding at individual sites suggest that stronger sequence signatures predict TF binding at previously inaccessible chromatin. Thus, our results support the argument that so-called pioneering abilities associated with a TF may not be absolute, but rather site-specific.

In future work, it will be of interest to examine how the relative contributions of sequence and preexisting chromatin vary in determining the binding of a wider range of TFs, and across a wider array of cell types. Identifying such sequence and chromatin

predeterminants of TF binding will be crucial for understanding gene regulation in various dynamic systems such as development and cellular programming.

## Methods

### ChIP-seq and ATAC-seq data (mEB cells)

Generation of the inducible iAscl1 and iNeurog2 mouse ES cell lines and corresponding ChIP-seq data is more completely described in Aydin et al. [42]. Briefly, inducible cell lines were generated using the inducible cassette exchange (ICE) method as previously described [59]. TF gene constructs are inserted in single copy into the expression-competent HPRT locus. The resulting iAscl1 and iNeurog2 ES cells are differentiated on untreated plates for 2 days to form embryoid bodies, and then expression of the transgene is induced via Doxycycline. Ascl1 and Neurog2 binding was assayed by ChIP-seq 12 h after Dox induction using the anti-Ascl1 (Abcam, ab74065) and anti-Neurog2 (Santa Cruz, SC-19233) antibodies. We assayed histone modifications as well as chromatin accessibility in EBs with ChIP-seq and ATAC-seq, respectively (Additional file 2: Table S1). We collected additional publicly available histone modification and histone variant ChIP-seq datasets from mouse ES cells (Additional file 2: Table S2). Together, our dataset defining the chromatin environment of mouse pluripotent cells consists of the following 12 data types: ATAC-seq, H2A.Z, acH2A.Z, H3K27ac, H3K27me3, H3K9me3, H3K4me1, H3K4me2, H3K4me3, H3K9ac, H4K20me3, and H3K36me3.

### ChIP-seq and ATAC-seq data (NIH-3T3 cells and BJ cells)

ChIP-seq data for TF inductions in mouse NIH-3T3 fibroblasts was retrieved from Raccaud et al. (GSE119784) [47]. We filtered for TFs that were not expressed as defined by RNA-seq in the NIH-3T3 cell line [47]. We used NCIS to estimate the sequenced control-based normalization factors for each TF ChIP-seq experiment [60]. Further, we filtered out induced TFs that had a MultiGPS-reported signal fraction < 0.01 and were single-replicate ChIP-seq experiments (Additional file 2: Table S3). We used five ATAC-seq experiments (Additional file 2: Table S4) as replicates to construct the network's ATAC-seq input [47]. ChIP-seq data for FOXA2, GATA4, and OCT4 inductions in human BJ fibroblasts were retrieved from Donaghey et al. (GSE110214) [23]. ATAC-seq, H3K27me3, H3K4me2, and H3K27ac were used to construct the network's chromatin input.

### TF ChIP-seq and preexisting chromatin data processing

Fastq files were aligned to the mouse (version mm10) or human (version hg38) genomes as appropriate using Bowtie (version 1.0.1) [61] with options "-q --best --strata -m 1 --chunkmbs 1024". Only uniquely mapped reads were considered for further analysis. MultiGPS [62] (version 0.74) was used to define transcription factor DNA binding events and was run with default options except for "--fixedpb 5" and excluding ENCODE black-list regions. A $q$ value cutoff of 0.01 (assessed using binomial tests and Benjamini-Hochberg multiple hypothesis test correction) was used to call statistically significant binding events with respect to sequenced input material collected from the same cell line. Peak-finding statistics are reported in Additional file 2: Table S3. Paired-end ATAC-seq reads were aligned using Bowtie2 (version 2.2.2) using the "-q --very-sensitive" options [63]. ChromHMM [51] (version 1.2.0) was run using default parameters.

### Training and test set construction

For testing, we divided the genome into 500 bp non-overlapping windows. For training, we use 500 bp overlapping windows, each of which are sequentially offset by 50 bp. Genomic windows overlapping peak calls with a $p$ value $\leq 0.001$ are labeled as bound. Windows overlapping non-significant peaks from MultiGPS are labeled ambiguous. All other genomic windows ($\sim 99\%$) are labeled as unbound. The sequence sub-network (Bichrom$_{SEQ}$) takes as input 500 bp sequences. Each nucleotide is encoded as a one-hot vector, such that only the index corresponding to the input nucleotides is set to one, and all other indices are set to zero. For each chromatin input data track, we extract the per-base read counts at each genomic locus. These raw coverage counts are binned into ten 50 bp non-overlapping bins (covering 500 bp windows). The binned read counts are total tag normalized for each replicate, and we use the replicate average at each bin as input to our network. The chromatin datasets are stacked, resulting in a 10 x $k$ chromatin input, where $k$ is the number of assayed histone modifications/chromatin accessibility. For analyses using only prior ATAC-seq (i.e., iAscl1 + 12 h, iNeurog2 + 12 h, and NIH-3T3 induced TFs). $k = 1$. For analyses of the mEB-induced TFs using prior ATAC-seq and other histone modification data, $k = 12$. For analyses of human BJ fibroblast-induced TFs, $k = 4$.

### Neural network architecture

The bimodal network architecture was designed to produce interpretable latent embeddings. We note that our bimodal architecture generally fits with the recently described concept of a Neural Additive Model [64], which linearly combines neural networks that each attend to a single input feature or feature type. Since hybrid CNN-LSTM networks have previously been shown to perform well at TF binding prediction tasks, we chose to use a CNN-LSTM based architectures for Bichrom$_{SEQ}$ and Bichrom$_{CHR}$ [24, 48]. The parameters of the individual sub-networks (Bichrom$_{SEQ}$ and Bichrom$_{CHR}$) were selected using a limited hyper-parameter random grid search (chr10 held-out test set, chr17 validation). We used a random grid-search over the (1) number of dense layers, (2) size of the dense layers, (3) number of convolutional filters, (4) activation functions, and the (5) dropout rate (see tested parameter values in Additional file 2: Table S6). To test whether our selected CNN-LSTM based architecture performed at par with alternative CNN-based architectures (e.g., deeper multi-layer CNNs), we generated a range of network architectures by randomly selecting hyper-parameter value combinations over (1) number of convolution layers, (2) convolutional kernel size, (3) number of convolutional filters, (4) max pooling size, (5) max pooling stride, (6) number of dense layers, (7) number of dense nodes, and (8) dropout rate to select network hyper-parameters (see tested parameter values in Additional file 2: Table S7). We found that Bichrom's CNN-LSTM network performed comparably to architectures with multiple convolutional layers when predicting Ascl1 binding in mEB cells (Additional file 1: Fig. S1).

In the Bichrom$_{SEQ}$ sequence sub-network, the 500 bp, one-hot encoded sequence input is first subjected to a 1-dimensional convolution layer, with each index in the one-hot encoding acting as a channel into this convolution. The convolutional layer consists of $240 \times 20$ bp long filters. The convolutional filters within 15 bp intervals are max-pooled, and the pooled convolutional output is used as input into a long short-term memory (LSTM) layer. The LSTM outputs a 32-vector, which then passes through two

dense layers, both subjected to ReLU activation and dropout. The activations from the final dense layer are input into a single tanh activated dense node. The Bichrom$_{CHR}$ chromatin sub-network uses convolutional filters that span two input bins. The filters are followed by an LSTM to model any observable tag densities discriminative of TF binding. The LSTM activations are input into a single dense layer followed by a single tanh activated dense node. The activations of both sub-networks are weighted by a final sigmoid activated node, used to output binding probability. The network is trained to predict ChIP-seq by minimizing the binary cross-entropy loss $J$:

$$ J = -\frac{1}{N} \sum_{i=1}^{N} y_i \log(\widehat{y_i}) + (1 - y_i) \log(1 - \widehat{y_i}) $$

Area under the precision-recall curve is used as a metric to measure network performance. Chromosome 17 is held out as a validation chromosome. Chromosomes 10–16 and chromosomes 18–19 are sequentially held out as test chromosomes in a *k*-fold training procedure.

### Neural network training strategies

To prevent the sequence-only network from learning accessibility-related sequences, we customize the sampling used to construct mini-batches for gradient descent-based training. We construct training batches such that in each batch, either only accessible or only inaccessible bound and unbound training data is used. This sampling strategy reduces model false positives at preexisting accessible regions, and leads to an improvement in sequence-only model performance measured on held-out chromosomes, as measured via the area under the precision recall curve (auPRC) (Additional file 1: Fig. S4A, B). For TFs with less than 10% of their binding sites in previously inaccessible chromatin, a chromatin-matched training results in poor predictive performance due to the small number of positive training examples in previously inaccessible chromatin. For these TFs, we use a less stringent strategy wherein within each batch, the fraction of previously accessible sites is similar in both the bound and unbound training instances.

The bimodal network aims to learn both sequence and prior chromatin signatures that characterize genome-wide TF binding. We can thus no longer control for accessibility distributions across bound and unbound training sets to prevent spurious learning of prior chromatin-related sequence signatures when training the bimodal network. To address this problem, we transfer weights from the previously trained sequence-only network to the Bichrom$_{SEQ}$ sequence sub-network in Bichrom's bimodal network. While the lower-level layer sequence sub-network weights are kept fixed during Bichrom training, the weights for the final dense layer in the Bichrom$_{SEQ}$ sub-network are allowed to vary to fit the genome-wide TF binding data.

Keeping the convolutional kernels fixed while re-training the final dense layers should allow the network to optimally predict TF binding, without learning new accessibility-related sequence features [65]. However, to ensure that the compensatory sequence-chromatin behavior observed across TFs is not due to such a network parameterization, we train a control network in which all dense weights within the Bichrom$_{SEQ}$ sub-network are kept fixed. We find that the 2-dimensional embeddings for each TF

binding site retain the same compensatory pattern as observed earlier, suggesting that network parameterization is not responsible for the observed sequence-chromatin compensation (Additional file 1: Fig. S5). Bichrom's joint bimodal network with fixed convolutional kernels can therefore be trained using imbalanced batches constructed by random sampling unbound data across the genome.

### Benchmarking against the ENCODE DREAM challenge data

The ENCODE DREAM challenge data was downloaded from https://www.synapse.org/#!Synapse:syn6131484/wiki/402026. The chromosomes available for training by the challenge were chromosomes chr2–7, chromosomes 9–20, and chromosomes 22 and X. Methods described in the ENCODE-DREAM challenge were tested on chromosomes 1, 8, and 21. As the challenge held-out data has not been made available at time of writing, we additionally held-out chromosome 18 from our training set in order to test our models. We used DNA sequence and concurrent DNase-seq as inputs into the bimodal network. We used the window definitions derived from the challenge (200 base pair windows, 50 base pair step size) in order to be consistent with other methods in the challenge. Notably, we did not make use of concurrent RNA-seq data or in vitro PBM-derived TF motif data that was used by several other methods. We evaluated network performance using the auPRC and compared our results to those reported for the top-performing methods in the challenge (Additional file 1: Fig. S2).

### Deriving the latent network embeddings

Let $\phi_S$ and $\phi_C$ represent the non-linear transformations applied to sequence feature vectors $X_S$ and chromatin feature vectors $X_C$ respectively. Then, for each input genomic window $i$, $\phi_S(X_S^i)$ and $\phi_C(X_C^i)$ represent real-valued outputs of the sequence and chromatin sub-networks at that window.

The bimodal network models the network output $y$ as follows:

$$\text{logit}(y) = \beta_0 + \beta_S \phi_S(X_S) + \beta_C \phi_C(X_C)$$

Here, $\beta_S$ and $\beta_C$ are weights assigned to sequence and chromatin sub-networks respectively, and $\beta_0$ is the bias term. Due to the linear relationship between $logit(y)$ and the transformed predictors, the weighted sub-network activations can be interpreted as the relative contributions of each modality to the networks output probability at a window $i$:

$$\text{seqscore}(i) = \beta_S \phi_S(X_S^i) \in \mathbb{R}$$

$$\text{chromscore}(i) = \beta_C \phi_C(X_C^i) \in \mathbb{R}$$

Therefore, the network embeds each input data point $i$ in a 2-dimensional space defined by seqscore($i$) and chromscore($i$).

### Feature attribution with integrated gradients

We use integrated gradients [49] (IG) to estimate the relative importance of each nucleotide $(x_i)_{i=1}^L$ within each input sequence $x$ of length $L$ base pairs. Due to the non-linearity in neural networks, it is not trivial to assign relative feature importance using the feature-associated network weights. Instead, gradient-based techniques are often

used in order to estimate relative feature importance. Vanilla gradients estimate feature importance as the partial derivative of the network output with respect to each input feature (in our case nucleotide $x_i$). However, due to the presence of multiple saturating functions used in neural networks, vanilla gradients do not satisfy the property of sensitivity. In other words, they are not guaranteed to assign non-zero attribution to all features $x_i$ that alter the network output when compared to reference feature vectors that inherently produce zero output probabilities. Integrated-gradients considers how predictions at input feature vectors differ from reference feature vectors. More specifically, integrated gradients calculate the gradients at all points along a straight-line path from the reference feature vector $x^b$ to the input feature. In our case, we define a reference feature as a sequence vector such that at each position, each nucleotide is equally likely. In other words, our reference sequence is a 4 * 500 matrix, with each column defined as [0.25, 0.25, 0.25, 0.25]. We implemented integrated gradients as defined in Sundarajan et al. [49]

### Analysis of sequence and preexisting chromatin predictors of Ascl1 binding sites

As described above, we applied integrated gradients (IG) to identify the sequence motifs driving sequence sub-network scores for Ascl1 [49]. However, attribution methods typically operate on individual sequences. To overcome the local behavior of such attribution methods, we extracted 20 bp sequences surrounding the local IG peaks; i.e., we extracted regions or "hills" that drive the neural network output from each sequence bound by Ascl1. We clustered the IG-derived hills based on their underlying sequence features using a K-means based clustering procedure described previously in SeqUnwinder [66]. Briefly, only $k$-mers present in at least 5% of IG-derived hills are used for clustering, and K-means clustering is performed using Euclidean distance as a metric [66]. We then perform motif discovery using MEME [67] to identify the enriched motifs within each K-means defined cluster [66].

To investigate the correlation between Bichrom$_{SEQ}$ scores and Ascl1 cognate motif multiplicity, we divided bound loci into categories based on their motif multiplicity and measured the Bichrom$_{SEQ}$ sub-network scores for each category. Only 5% of all Ascl1-bound sites lacked exact matches to the core Ascl1 E-box motif CAGSTG, and these sites were assigned the lowest median scores by the sequence sub-network (Fig. 4c). To ensure that increased sequence network scores at Ascl1 binding sites with multiple motifs do not stem from the presence of confounding features at these sites, we directly tested the relationship between motif multiplicity and Bichrom$_{SEQ}$ scores. We inserted between one and five randomly spaced CAGSTG $k$-mers in a set of 10,000 randomly generated 500 bp sequences (background frequencies A/T = 0.5 and G/C = 0.5). We divided the simulated 500 bp sequences into categories based on the number of embedded CAGSTG $k$-mers. Each simulated sequence was input into a Bichrom network trained using Ascl1 ChIP-seq data and the distribution of Bichrom$_{SEQ}$ sub-network scores at each sequence category was calculated (Fig. 4d).

To investigate the effect of various nucleotides flanking the Ascl1 cognate motif, we constructed a sequence in which each position is encoded as a $[0.25, 0.25, 0.25, 0.25]^T$ vector; i.e., each base $[A, T, G, C]^T$ occurs with equal probability at each position in this

sequence. We inserted CAGSTG $k$-mers flanked by variable nucleotides into this reference sequence and again scored them with the Ascl1-trained Bichrom$_{SEQ}$ sub-network.

To investigate the relationships between Bichrom$_{CHR}$ scores at Ascl1 binding sites and individual preexisting chromatin features (Fig. 5a), we divided Ascl1 binding sites into quartiles based on their Bichrom$_{SEQ}$ scores and calculated the mean normalized tag enrichment for each preexisting chromatin feature at each quartile. Further, we identified genome-wide enrichment domains using the DomainFinder utility in Seq-Code (https://github.com/seqcode/seqcode-core/blob/master/src/org/seqcode/projects/seed/DomainFinder.java) and calculated Bichrom$_{CHR}$ scoring distributions at all preexisting chromatin domains for each input chromatin dataset (Fig. 5b). We also calculated the distribution of Bichrom$_{SEQ}$ and Bichrom$_{CHR}$ scores at Ascl1 binding sites categorized into 12 states based on ChromHMM state calls (Fig. 5c).

### Motif discovery

De novo motif discovery for iAscl1 induced Ebf2, Onecut2, and Brn2, the mouse NIH-3T3 induced TFs and the human BJ induced TFs was performed using MEME-ChIP [68] (version 5.1.0) with default settings. All motifs with MEME E-values < 0.01 discovered at SP and CP sites were reported. Repetitive poly-A or poly-T repeats were excluded. Motif scanning was performed using FIMO [69], all hits greater than the default $p$ value threshold of 1e−4 were reported.

### The posterior distribution of the model recall

We used the model recall at a fixed false positive rate (FPR) to compare model performance across TFs. TPs are true positives in the held-out test set, whereas FNs are false negatives in the test set.

$$\text{Recall} = \frac{\text{TPs}}{\text{TPs} + \text{FNs}}$$

However, we note that ChIP-seq signal fractions and the number of peaks called vary widely across TFs. Models trained to predict binding for TF ChIP-seq experiments that contain smaller numbers of peaks (and correlated lower signal fractions) suffer from having access to limited training data. In order to quantify our confidence in the model recall, we use a probabilistic framework that models the recall for each TF given the underlying ChIP-seq data. Specifically, analogous to Brodersen et al. [70], we consider the observed model recall (measured on a single held-out test chromosome) to be an actualization of an underlying true recall value $r$ given $N$ independent Bernoulli trials, where $N$ is the number of binding sites in the held-out test chromosome. Each binding site can be either labeled a true positive (success) or a false negative (failure) by the network.

$$\text{Recall} \sim \text{Binomial}(N, r)$$

We derive the posterior distribution of the recall $r$ assuming a beta prior (parameters $a = 1$, $b = 1$; equivalent to a uniform prior—for details, see Brodersen et al. [70]). The mode of this posterior distribution is the observed model recall. If a TF ChIP-seq experiment contains a small number of peaks, the distribution of $r$ has high variance (e.g.,

Brn2, Fig. 6a). On the other hand, a low variance in the distribution of $r$ reflects a higher degree of confidence in our estimate of the recall (e.g., Ebf2, Onecut2, Fig. 6a).

## Supplementary Information

---

**Additional file 1.** Integrated supplementary figures. Contains supplementary figures from S1 to S12.

**Additional file 2.** Supplemental data and data source descriptions. Contains supplementary tables from S1 to S7 and all data source citations.

**Additional file 3.** Review history.

---

### Acknowledgements
We thank Chunyu Ma for a critical reading of the manuscript. We also thank Dr. William Lai for helping us test the Bichrom software.

### Peer review information

### Review history
The review history is available as Additional file 3.

### Authors' contributions
D.S. conceived of the computational application, with guidance from S.M. D.S. set up the computational framework and design. B.A. performed the ChIP-seq and ATAC-seq experiments, with guidance from E.O.M. D.S. and S.M. co-wrote the manuscript. All authors read and approved the final manuscript.

### Authors' information
Twitter handles: @divyanshi91 (Divyanshi Srivastava); @BegumAydin__ (Begüm Aydin); @EstebanMazzoni (Esteban O. Mazzoni); @mahonylab (Shaun Mahony).

### Funding
This manuscript is based upon work supported by the National Science Foundation ABI Innovation Grant No. DBI1564466 (to S.M.). Any opinions, findings, and conclusions or recommendations expressed in this material are those of the authors and do not necessarily reflect the views of the NSF. This work was also supported by the Academic Computing Fellowship (to D.S.), NIGMS R01GM121613 (to S.M.), NICHD R01HD079682 (to E.O.M.), and an NVIDIA GPU equipment grant.

### Availability of data and materials
Ascl1, Brn2, Ebf2, and Onecut2 ChIP-seq data have been uploaded to GEO under accession GSE114176 [42]. The mouse EB chromatin datasets were downloaded from GSE80483 [54]. Additional mouse ES chromatin ChIP-seq datasets were downloaded from GSE39237 [71], GSE49847 [72], and GSE12241 [73]. TF ChIP-seq and ATAC-seq data in mouse NIH-3T3 cells were downloaded from GSE119784 [47]. TF ChIP-seq and ATAC-seq data in human BJ fibroblasts were downloaded from GSE90456 [23]. Open source code (MIT license) for Bichrom is available for download on github: https://github.com/seqcode/Bichrom [74]. A release of the Bichrom source code (v0.1.2) is available on Zenodo: https://doi.org/10.5281/zenodo.4271912 [75]. A detailed list of data sources and code repositories has been provided in Additional file 2.

### Ethics approval and consent to participate
Not applicable

### Consent for publication
Not applicable

### Competing interests
The authors declare that they have no competing interests.

### Author details
[1]Center for Eukaryotic Gene Regulation, Department of Biochemistry & Molecular Biology, Pennsylvania State University, University Park, PA, USA. [2]Department of Biology, New York University, New York, NY, USA.

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

## 