## [**Additional file 3.** Review history. · Genome Biology]

Review History

First round of review

Reviewer 1

Are you able to assess all statistics in the manuscript, including the appropriateness of statistical tests used? Yes, and I have assessed the statistics in my report.

Comments to author:

In this paper, Srivastava et al. studied how pre-existing chromatin environment (e.g., accessibility, histone modifications, etc.) can influence transcription factor (TF) binding at sequence specific sites. Their motivation is to discover the role of cell-type specific chromatin environment in establishing cell-type specific binding of TFs. As the authors have discussed, there are prior works on modeling genome wide TF binding from sequence and cell-type specific "concurrent" chromatin environment. However, they argue that this approach cannot discern which aspects of chromatin environment can induce TF binding rather than being a "parallel measurement" of TF binding itself. Therefore, the authors chose to model datasets that have assessed both prior chromatin environment and TF occupancy upon overexpression. Using a bimodal (one mode for sequence, the other mode for chromatin environment) neural network, the authors find that sequence specificity and chromatin environment have compensatory roles in overall TF binding. Variation in both of these inputs leads to variation in binding. Their models also suggest TF binding is positively influenced by multiplicity of sequence motif occurrences.

This is an important mechanistic question, but it is not possible to establish a causal link between chromatin environment and TF binding with only two datasets. In particular, one needs to perform a number of controlled knock-outs to elicit causal links. The authors should make this clear that their motivating question and the question being answered are different. There are several concerns about their models and modeling approach (see below). But the major concern is, these new models did not discover any novel mechanism/hypothesis. Our other comments and concerns are as follows.

1. TFs that are already bound to the DNA can also impact binding of a TF when it is overexpressed. How do the authors deal with this fact?
2. Except for the choice of appropriate datasets, it is not clear whether and how the bimodal deep neural network is a novel contribution. It is arguable that the previous models (for modeling TF binding from sequence and "concurrent" chromatin environment) should provide similar conclusions if applied on these new datasets.
3. It is also not clear why the authors needed a deep neural network. How was this architecture selected? What type of sequence patterns did the 240 convolutions discover? Do they discover motifs of any potential co-factor? Also, we do not find any indication of what they achieved by including the LSTM layers. Typically, these recurrent layers should discover some "grammar" in the sequence space, but the authors only found some role of motif multiplicity.
4. The parameters in the final dense layer of the sequence sub-network were not kept fixed when training the combined model. How can the authors rule out the possibility that the compensatory contributions from sequence and chromatin environment is not an artifact of model reparameterization?

Minor comments:

1. Pg. 8: "Orthogonal strategy to confirm that ... binding sites" -> should be "binding peaks"?
2. Please give a more detail description of the feature attribution method.
3. Regarding the posterior distribution of recall: this estimate may still be biased depending on which

chromosome was held-out and the distribution of peaks across the chromosomes. Also, the authors should note that, for beta with $a = 1$, $b = 1$, it is simply a uniform distribution.

Reviewer 2

Are you able to assess all statistics in the manuscript, including the appropriateness of statistical tests used? Yes, and I have assessed the statistics in my report.

Comments to author:

- Are the methods appropriate to the aims of the study, are they well described, and are necessary controls included?

1. Please clarify where your data comes from. It seems to me that all your data were from mouse cell lines, so claiming that your model provides a framework for modeling chromatin landscapes *in vivo* is a stretch. Induction of iAscl1 and iNeurog2 ES cells on a plate is considered as *in vitro* experiments, induction of those cassettes in live mice and then harvesting the cells for ChIPseq analysis is *in vivo*.

2. Incorporating prior chromatin information does lead to an increase of specificity of induced Ascl1 binding predictions, however, the median auPRC for both sequence and preexisting chromatin combined is only at 0.59. Is the improvement in prediction significant? Is the sequence+preexisting chromatin model accurate enough to predict TF binding?

- Are the conclusions adequately supported by the data shown?

3. Your network does indeed rely on motif multiplicity and motif flanks in predicting Ascl1 binding, however simulated sequences is not enough to validate the effects of motifs on TF binding.

4. Your conclusions for other TFs are unclear. More details on your analysis and results is needed for the last section.

- Are sufficient details provided to allow replication and comparison with related analyses that may have been performed?

Yes.

- Does the work represent a significant advance over previously published studies?

Yes.

- Is the paper of broad interest to others in the field, or of outstanding interest to a broad audience of biologists?

Yes, temporal analysis of TF binding is very important in understanding TF's complex binding specificity. The authors examined the determinants for newly activated TF binding in mouse cell lines (*in vitro*).

Authors Response

Responses to reviews for GBIO-D-19-00981

An interpretable bimodal neural network characterizes the sequence and preexisting chromatin predictors of induced TF binding

(previously titled: "Characterizing the sequence and prior chromatin determinants of induced TF binding with bimodal neural networks")

We thank all the reviewers for their detailed comments, which have led us to improve and clarify our manuscript in several places. In what follows, the reviewer's comments are given in **black** text, interleaved with our responses in **blue**. We have extensively revised the entire manuscript in response to the reviewer comments, and to further improve aspects beyond the reviewer comments. Thus, we do not provide a "tracked changes" manuscript or show individual edits below.

Reviewer #1

In this paper, Srivastava et al. studied how pre-existing chromatin environment (e.g., accessibility, histone modifications, etc.) can influence transcription factor (TF) binding at sequence specific sites. Their motivation is to discover the role of cell-type specific chromatin environment in establishing cell-type specific binding of TFs. As the authors have discussed, there are prior works on modeling genome wide TF binding from sequence and cell-type specific "concurrent" chromatin environment. However, they argue that this approach cannot discern which aspects of chromatin environment can induce TF binding rather than being a "parallel measurement" of TF binding itself. Therefore, the authors chose to model datasets that have assessed both prior chromatin environment and TF occupancy upon overexpression. Using a bimodal (one mode for sequence, the other mode for chromatin environment) neural network, the authors find that sequence specificity and chromatin environment have compensatory roles in overall TF binding. Variation in both of these inputs leads to variation in binding. Their models also suggest TF binding is positively influenced by multiplicity of sequence motif occurrences.

This is an important mechanistic question, but it is not possible to establish a causal link between chromatin environment and TF binding with only two datasets. In particular, one needs to perform a number of controlled knock-outs to elicit causal links. The authors should make this clear that their motivating question and the question being answered are different. There are several concerns about their models and modeling approach (see below). But the major concern is, these new models did not discover any novel mechanism/hypothesis. Our other comments and concerns are as follows.

We thank the reviewer for their feedback, and we agree with their points here. We have extensively rewritten our manuscript to emphasize that our work provides a computational framework that can be used to generate hypotheses about the relationships between TF binding and prior chromatin landscapes. We have also removed language suggesting that the links found by our approach are necessarily causal. We now make it clear that our computational approach – Bichrom – aims to find sequence and prior chromatin predictors that explain observed TF binding sites. Such predictors can suggest hypotheses that would have to be tested before causal links are established.

While our revised manuscript now more clearly focuses on the goal of interpreting observed TF binding in the context of sequence and prior chromatin predictors, we also wanted to demonstrate that the hypotheses suggested by Bichrom can indeed capture causal links between the preexisting chromatin accessibility environment and induced TF binding. The reviewer rightly suggests that causality sometimes can be established with knock-out experiments, but this is not always simple for the types of temporal associations that Bichrom suggests. For example, it is difficult to think of a knock-out experiment that would confirm a causal link between general chromatin accessibility and future TF binding. However, the same hypothesis can be tested by using Bichrom to characterize the predictors of induced TF binding in one chromatin state and testing whether the conclusions hold up in a different chromatin environment.

In newly added work, we analyzed ChIP-seq data from 3 transcription factors – Brn2, Ebf2, and Onecut2 – that become expressed in two distinct chromatin environments. Specifically, we use Bichrom to predict the relative degree to which each TF's binding sites are predicted by prior chromatin accessibility. Importantly, we base this Bichrom analysis on data from only one of the two chromatin environments in which we characterized Brn2, Ebf2, and Onecut2 binding. We find that prior accessibility adds a significant improvement in predictive capacity for 2 of the 3 TFs compared with using sequence information alone, but no significant improvement is observed for Onecut2. In our interpretation, Bichrom's results suggest that Onecut2 may be less sensitive to the prior chromatin environment than Brn2 or Ebf2. To test this hypothesis, we examined the 3 TFs' binding in the alternate chromatin environment. Supporting our interpretation, Onecut2 binding changes much less than the others across the alternate prior chromatin contexts, suggesting that it is less sensitive to the prior chromatin landscape (**Fig. 6A-B, Supp. Fig. 6 and Supp. Fig. 7**). Furthermore, individual Brn2, Ebf2, and Onecut2 sites that Bichrom predicts as "chromatin-predicted" or "sequence-predicted" show higher or lower (respectively) variability in binding in the alternate chromatin environment (**Fig. 6C-F**).

These results are more fully described in the new Results section titled "**Bichrom predicts the relative dependence of neuronal TF binding sites on preexisting chromatin**", and add an important validation of our approach and interpretations.

1. TFs that are already bound to the DNA can also impact binding of a TF when it is overexpressed. How do the authors deal with this fact?

Because chromatin states largely depend on TFs, we agree that TFs which are already bound in a cell type influence the future binding of TFs that become expressed in that cell type. Our method, Bichrom, can be used to explore such associations in several ways. If the user has ChIP-seq data available for a particular TF that they believe may impact future TF binding, that ChIP-seq data can be incorporated into Bichrom alongside the other prior chromatin predictors. Then Bichrom's results can explicitly address the degree to which an induced TF's binding is predicted by the preexisting TF. On the other hand, if TF ChIP-seq data is not available, the binding activities of preexisting TFs are encoded within other chromatin feature tracks (e.g. accessibility or enhancer-associated histone modifications). If we find that those features predict future TF binding, we naturally must be cautious with our interpretations. For example, an induced TF's binding may not

be causally dependent on prior accessibility, but rather on interactions with a preexisting TF that is itself causing the accessibility.

Our revised manuscript more clearly acknowledges such issues involving interpretation and possible interactions with pre-bound TFs. Further, in the revised manuscript, we have included additional analysis of latent network embeddings to identify secondary TF motifs that may relate to pre-bound TFs at both pre-inaccessible and pre-accessible chromatin. We demonstrate that we are able to find instances where secondary motif enrichment predicts future TF binding at a subset of prior-chromatin-predicted TFs binding sites for Ascl1, Ebf2, and Brn2 (**Fig. 4A, Fig. 6E, F**).

2. Except for the choice of appropriate datasets, it is not clear whether and how the bimodal deep neural network is a novel contribution. It is arguable that the previous models (for modeling TF binding from sequence and "concurrent" chromatin environment) should provide similar conclusions if applied on these new datasets.

The Reviewers' comment made us realize that we failed to communicate the novelty of the bimodal neural network framework. We have rewritten the manuscript to highlight the novelty of our framework, and we included additional data to demonstrate its validity. We discuss this in brief here.

The majority of previously described methods that integrate sequence and concurrent chromatin data to predict genome-wide TF binding use an early-integration framework, i.e. sequence and chromatin data is integrated at the feature level, and is jointly modeled by a unified framework²⁻⁶. It is not clear how such methods can be applied to **deconvolve** the sequence and preexisting chromatin accessibility predictors of TF binding the level of individual TF binding sites. Notably, Arvey, *et al.* have previously described a late-integration framework within which separate SVMs are used to model TF binding as a function of DNA sequence and concurrent chromatin accessibility data². However, in Arvey, *et al.*, the weights of each input modality (sequence vs. concurrent chromatin accessibility) are not derived using the input training data, but rather a normalized sum of the sequence and chromatin SVM outputs is used to predict TF binding.

The novelty of our bimodal neural network is its design, which embeds TF binding sites in a latent 2-dimensional space (**Fig. 1A, Fig. 3A, Supp. Fig. 9A-C**). The coordinates of each TF binding site in this space represent the degree to which sequence features and prior chromatin features predict TF binding at individual sites. **To our knowledge, Bichrom is the first method that quantifies the contributions of multiple modalities to TF binding prediction at individual sites across the genome.** The types of applications demonstrated in our manuscript (e.g., examining compensatory sequence and prior chromatin predictors at individual sites) are not trivially enabled by applying previous TF binding prediction approaches to temporally separated data.

The novelty of our approach is now more clearly described in the revised manuscript.

3A. It is also not clear why the authors needed a deep neural network. How was this architecture selected?

We use a neural network because of the flexibility it provides while modeling multimodal data; it is more straightforward to integrate sequence and chromatin predictors in a unified framework with CNN-based neural networks than it would be with alternate machine learning approaches. We would like to emphasize that the innovation in our framework is not the use of neural networks, but rather the neural network design within which distinct sub-networks transform higher-dimensional sequence and chromatin data into a directly interpretable lower-dimensional representation. We have tried to make this motivation clearer in our revised manuscript.

The architecture of the bimodal network (Bichrom) was designed such that both the sequence (Bichrom_{SEQ}) and chromatin sub-networks (Bichrom_{CHR}) output single real-valued activations, which are then weighted by a sigmoid-activated logistic node. The architectures and hyper-parameters of the individual sub-networks (Bichrom_{SEQ} and Bichrom_{CHR}) were selected using a limited hyper-parameter random grid search. We used a random grid-search over the number of (1) number of dense layers, (2) size of the dense layers, (3) number of convolutional filters, (4) activation functions and the (5) dropout rate. We used the size of the convolutional kernels as previously defined by DeepBind (kernel size=24). This is now more clearly described in our methods.

3B. What type of sequence patterns did the 240 convolutions discover? Do they discover motifs of any potential co-factor?

The Reviewer raises an excellent point about interpreting the sequence features learned by our approach. While convolutional kernels in 1-layer convolutional neural networks can learn directly interpretable TF sequence motifs, it is much less clear whether convolutional filters are similarly interpretable in deeper convolutional neural networks (CNNs) or CNN-LSTM architectures. In fact, convolutional kernels often learn overlapping, non-independent patterns. Therefore, analyzing patterns learned within individual convolutional kernels may not be a meaningful representation of the sequence features identified by the network.

In the revised manuscript, we use gradient-based techniques to identify local regions in the input sequences that are important for the network's output predictions. We cluster these "important" regions based on their underlying *k*-mer frequencies, and perform motif discovery at these individual clusters (**Fig. 4A, 6D-F**). Our results demonstrate that in addition to motif multiplicity, secondary motif enrichment and flanking nucleotide composition can further influence the network output predictions.

3C. Also, we do not find any indication of what they achieved by including the LSTM layers. Typically, these recurrent layers should discover some "grammar" in the sequence space, but the authors only found some role of motif multiplicity.

The Reviewer is correct that the typical motivation behind the use of recurrent neural networks (e.g., LSTMs) is to introduce the ability to model spatial dependencies between convolutional filter activations. In the TF binding setting, it is tempting to assume that a CNN-LSTM architecture should learn motif "grammars" within enhancer regions, for example. Using our gradient-based feature importance approach, we did not find that Bichrom learned any spatial dependencies between motifs for the examined TFs. This could either be because the learning or interpretation schemes fail to detect such dependencies, or because such dependencies don't biologically exist for the examined TFs – we honestly can't tell at this point.

So, what is the point in continuing to use a LSTM layer? We believe that they are useful in introducing spatial dependencies *within* TF binding motifs. If individual convolutional filters learn *parts* of a motif (or informative motif-flanking features), LSTMs are useful in stitching them together at an individual binding site. It's worth noting that the same effect can likely be achieved using multiple stacks of convolutional layers, but using an LSTM allows us to use a shallower neural network architecture overall.

In the revised manuscript, we demonstrate that our chosen CNN-LSTM architecture performs better than, or on par with, a wide range of CNN-only architectures, including some that have much deeper numbers of layers. Specifically, we compare Bichrom's CNN-LSTM with CNN-based networks with combinations of selected values for the following parameters: (1) number of convolution layers, (2) convolutional kernel size, (3) number of convolutional filters, (4) max pooling size, (5) max pooling stride, (6) number of dense layers, (7) number of dense nodes and (8) dropout rate (**Supp. Fig. 1**).

4. The parameters in the final dense layer of the sequence sub-network were not kept fixed when training the combined model. How can the authors rule out the possibility that the compensatory contributions from sequence and chromatin environment is not an artifact of model reparameterization?

To address this concern, we trained the bimodal network (Bichrom) while keeping the final dense layer in Bichrom_{SEQ} fixed, demonstrating that network re-parameterization is not responsible for the observed pattern in which compensatory contributions from sequence and the prior chromatin environment predict TF binding (**Supp. Fig. 5**).

Minor comments:

1. Pg. 8: "Orthogonal strategy to confirm that ... binding sites" -> should be "binding peaks"?

This sentence has been removed in the revised manuscript.

2. Please give a more detail description of the feature attribution method.

We have added a more detailed description of the feature attribution method to the revised manuscript.

3. Regarding the posterior distribution of recall: this estimate may still be biased depending on which chromosome was held-out and the distribution of peaks across the chromosomes. Also, the authors should note that, for beta with $a = 1$, $b = 1$, it is simply a uniform distribution.

We agree that the recall estimates may be biased based on the chromosome being tested. In our revised manuscript, we have repeated training over 9 held-out test chromosomes for each TF. In addition to recall values, we report the auPRC for each training iteration. Further, to test whether the gain in auPRCs upon the addition of preexisting chromatin are significant, we compare distributions of auPRCs from the sequence-only network and Bichrom using the Wilcoxon Signed Rank test. Finally, we have added in a note in the manuscript that the beta(1,1) prior is equivalent to a uniform prior.

Reviewer #2

We thank the Reviewer for their insightful comments throughout.

Are the methods appropriate to the aims of the study, are they well described, and are necessary controls included?

1. Please clarify where your data comes from. It seems to me that all your data were from mouse cell lines, so claiming that your model provides a framework for modeling chromatin landscapes *in vivo* is a stretch. Induction of iAscl1 and iNeurog2 ES cells on a plate is considered as in vitro experiments, induction of those cassettes in live mice and then harvesting the cells for ChIPseq analysis is in vivo.

The Reviewer's comment reflects the varying uses of *in vitro* versus *in vivo* terminology. Some of the TF-DNA binding literature uses "*in vitro*" to refer to experiments carried out on naked DNA

templates, and “*in vivo*” to describe experiments performed within a cellular chromatin context (e.g. via ChIP-seq). We apologize for the confusion. As the reviewer pointed out, all of the ChIP-seq data we examined comes from cultured ES cells or fibroblast cell lines, and these are of course “*in vitro*” systems. To avoid confusion, we removed the “*in vivo*” terminology and more clearly describe the ES-based cellular systems used through much of the manuscript.

2. Incorporating prior chromatin information does lead to an increase of specificity of induced Ascl1 binding predictions, however, the median auPRC for both sequence and preexisting chromatin combined is only at 0.59. Is the improvement in prediction significant? Is the sequence+preexisting chromatin model accurate enough to predict TF binding?

We first address whether our bimodal architecture can accurately predict genome-wide TF binding (or put another way, is 0.59 a reasonable auPRC compared with other state-of-the-art methods?). Briefly, to evaluate the ability of our bimodal network (Bichrom) to accurately predict genome-wide TF binding, we turned to the ENCODE-DREAM TF binding challenge, which evaluated the ability of a range of computational methods to predict TF binding within and across cell types. The ENCODE DREAM challenge was set up to predict TF binding as a function of sequence, concurrent chromatin accessibility (DNase-seq) and concurrent gene expression (RNA-seq) data. We used the same training regime as required by the challenge to ensure our results were comparable to other models (e.g., predictions were made in overlapping 200 bp windows, with a 50 bp stride). Importantly, we note that the goal of the ENCODE-DREAM challenge (predicting TF binding using concurrent accessibility and expression) is not the same as our motivation – we are primarily interested in building a framework that can deconvolve sequence and prior chromatin predictors of TF binding. That said, Bichrom performs at par with the current state-of-the-art models on the ENCODE-DREAM challenge datasets, albeit with slightly lower predictive accuracies (**Supp. Fig. 2**). In fact, for CTCF bound in induced pluripotent stem cells and PC-3 cells, the bimodal network outperforms the overall top-performing method, Anchor⁶. The auPRCs for predicting TF binding using concurrent chromatin data as a feature range from **0.30-0.79** for 12 TFs tested (**Supp. Fig. 2**). Thus, we are confident that the auPRC of 0.59 observed for predicting induced Ascl1 binding is a reasonable performance level for current approaches. There is still room for improvement in this task, of course.

Next, we address whether the **gain** in auPRC for Ascl1 is **significant** upon addition of prior mEB chromatin features. In our revised manuscript, we train both the bimodal network (Bichrom) and a sequence-only network using 9 independent training sets, each corresponding to a distinct held-out chromosome as a test set. We use the Wilcoxon signed rank test to compare whether the means of the paired samples (sequence vs. bimodal sequence-chromatin model auPRCs for each training set) are different. The p-value for Ascl1 is 0.003, suggesting that the gain in auPRC is statistically significant (**Fig. 2A**). We also perform significance testing for all other TFs analyzed (**Fig. 7, Supp. Fig. 7**).

Finally, we now demonstrate that gain in network predictive capacity (i.e., gain in auPRC) upon incorporation of preexisting chromatin can be interpreted as reflecting the degree to which preexisting chromatin predicts induced TF binding. As described in the first response to Reviewer #1, we apply Bichrom analysis to interpret the binding sites of Brn2, Ebf2, and Onecut2 that become expressed in two distinct chromatin environments¹. We show that Bichrom's estimates of the degree to which preexisting chromatin accessibility predicts these TF's binding are correlated with the overall degree of differential binding observed across the two distinct chromatin environments (**Fig. 6A, B and Supp. Fig. 7**), suggesting that the gain in auPRCs reflects the relative degree to which a TF depends on the preexisting chromatin.

Are the conclusions adequately supported by the data shown?

3. Your network does indeed rely on motif multiplicity and motif flanks in predicting Ascl1 binding, however simulated sequences is not enough to validate the effects of motifs on TF binding.

We agree with the Reviewer that simulated data isn't sufficient to validate the effects of sequence motif composition on TF binding. The relevant part of the results (**Fig. 4D**) was one of several analyses we performed in order to tease apart the various features learned by the Bichrom_{SEQ} sub-network. Our revised manuscript more clearly describes all such features as "predictors" of induced binding. While such predictors suggest causal hypotheses that may be examined using experimental approaches, such experiments are beyond the scope of the current work.

4. Your conclusions for other TFs are unclear. More details on your analysis and results is needed for the last section.

We have performed a more detailed analysis of the features learned by Bichrom when predicting the binding sites of TFs induced in mouse NIH-3T3 fibroblasts. We also include results from additional TFs induced in human fibroblasts (**Fig. 7, Supp. Fig. 10, 11, 12**).

Are sufficient details provided to allow replication and comparison with related analyses that may have been performed?

Yes.

Does the work represent a significant advance over previously published studies?

Yes.

Is the paper of broad interest to others in the field, or of outstanding interest to a broad audience of biologists?

Yes, temporal analysis of TF binding is very important in understanding TF's complex binding specificity. The authors examined the determinants for newly activated TF binding in mouse cell lines (in vitro).

Comment

Response

Second round of review

Reviewer 1

Srivastava et al. revised their previous manuscript on a bimodal neural network that assesses the role of sequence and preexisting chromatin landscapes in the occupancy of newly activated TFs. The authors have addressed our previous comments to avoid causal statements. This has really clarified the exact scientific question Bichrom is designed to answer.

Major comments on the revised version:

1. However, the authors' discussion on our other concern about whether this model a novel contribution is not satisfactory. It's still not clear how the other studies provide less explanation. Previous studies did not look at temporally preceding data of histone modifications, but if given the same data, how would those models be any weaker?

The authors can check PMIDs 26291518 and 26818008. For example, PMID 26291518 noted: "In addition to sequence motif (SM), chromatin state (CS) and DNA structure (DS) features have been shown to be informative for TFBS identification ... the relative contributions of SM, CS, and DS features, both singly and in combination, towards TF binding prediction remains unclear ... To address these questions, we first examined the relationships between TF binding regions determined in genome-wide ChIP analysis of 40 TFs and 23 features including two SM, 11 CS, and 10 DS features ... TFs differ greatly in which features are significantly different between bound and unbound regions. Taking RAP1 as an example, values for most of the CS features are significantly different between bound and unbound regions, yet none of the DS features is significant ... In contrast, for ZAP1, many DS but few CS-related features have significant test statistics ... However, some TFs, such as MSN2, REI1, SPT23, and SWI5, show significant differences both in CS and DS features". The other paper, PMID 26818008, also noted similar conclusions.

Overall, the authors need to do a better comparison against other highly similar works and show how the new tool is more powerful or completely novel than the published works.

The authors have mentioned Arvey et al.'s work. They are right that Arvey et al. have done things differently, it doesn't mean that the conclusions would be different.

2. In the new presentation, the authors have focused on the role of existing histone environment in a newly expressed TF. They're using chromatin marks at a time T0 in a model to predict ChIP data of a TF at a later time T1. It does not, however, show the role (if any) of chromatin marks at time T1 in the occupancy of the TF at T1. For a pioneer TF like Ascl1, this is an important part of Ascl1's binding mechanism.

3. The authors noted "Thus, our approach may offer a metric for quantifying the "pioneering" activity of a

TF; those that display less dependence on preexisting chromatin may be more likely to override those landscapes." Did they mean pioneer TFs to display less dependence on preexisting chromatin or the opposite? Ascl1 is a pioneer TF, but their model showed a major dependence on preexisting chromatin.

4. From Fig 7, are the models useful where auPRC were very bad?

Minor comment:

The paper "Chromatin Landscape Dictates HSF Binding to Target DNA Elements" by Guertin, M. J. & Lis, J. T. appears twice in the bibliographic references. The authors need to check if this happened with any other cited manuscript.

Reviewer 2

The authors have addressed my concerns and have included additional analysis that enhanced their work. Their new analysis on three other TFs added to the validity of their work. I believe their bimodal neural network framework is novel and crucial in deciphering the sequence and chromatin environmental factors contributing to new "induced" TF binding. Additionally, they've clarified the confusions the other reviewer and I previously had.

I suggest including a short descriptive title for each figure. Overall, I am satisfied with the revised manuscript. Good job.

Responses to reviews for GBIO-D-19-00981

An interpretable bimodal neural network characterizes the sequence and preexisting chromatin predictors of induced TF binding

Reviewer #1

Srivastava et al. revised their previous manuscript on a bimodal neural network that assesses the role of sequence and preexisting chromatin landscapes in the occupancy of newly activated TFs. The authors have addressed our previous comments to avoid causal statements. This has really clarified the exact scientific question Bichrom is designed to answer.

We thank the reviewer for recognizing the improvements in our manuscript, and for their additional comments below.

Major comments on the revised version:

1. However, the authors' discussion on our other concern about whether this model a novel contribution is not satisfactory. It's still not clear how the other studies provide less explanation. Previous studies did not look at temporally preceding data of histone modifications, but if given the same data, how would those models be any weaker?

The authors can check PMIDs 26291518 and 26818008. For example, PMID 26291518 noted: "In addition to sequence motif (SM), chromatin state (CS) and DNA structure (DS) features have been shown to be informative for TFBS identification ... the relative contributions of SM, CS, and DS features, both singly and in combination, towards TF binding prediction remains unclear ... To address these questions, we first examined the relationships between TF binding regions determined in genome-wide ChIP analysis of 40 TFs and 23 features including two SM, 11 CS, and 10 DS features ... TFs differ greatly in which features are significantly different between bound and unbound regions. Taking RAP1 as an example, values for most of the CS features are significantly different between bound and unbound regions, yet none of the DS features is significant ... In contrast, for ZAP1, many DS but few CS-related features have significant test statistics ... However, some TFs, such as MSN2, REI1, SPT23, and SWI5, show significant differences both in CS and DS features". The other paper, PMID 26818008, also noted similar conclusions.

Overall, the authors need to do a better comparison against other highly similar works and show how the new tool is more powerful or completely novel than the published works.

The authors have mentioned Arvey et al.'s work. They are right that Arvey et al. have done things differently, it doesn't mean that the conclusions would be different.

The reviewer points to three previous publications describing machine learning applications that use sequence and chromatin features to interpret TF binding sites: Tsai, *et al.* (PMID 26291518); Kumar, *et al.* (PMID: 26818008); and Arvey, *et al.* (PMID: 22955984). As acknowledged by the reviewer, all three of these publications focus on examining how concurrent chromatin features are correlated with TF binding sites in the same cell type, whereas our study uniquely examines the relationship between preexisting chromatin features and the future binding activities of an induced TF. However, the reviewer asks whether the same types of models described in these previous publications could be applied to the

data described in our manuscript, and if so, how our Bichrom approach would provide novel or additional insight. We appreciate the opportunity to clarify the novel aspects of our approach, both here and in the manuscript.

Directly comparing the performance of our approach to the models in the three previous publications turned out to be not as trivial as we had hoped. None of the three methods are available for download (neither Tsai, *et al.* nor Kumar, *et al.* provided any code, whereas Arvey *et al.* only provided code for the part of their approach that trains on sequence features). Furthermore, none of the three publications focused on predicting/explaining TF binding patterns across entire mammalian-scale genomes, whereas Bichrom does. The methods described in the three publications (Random Forests and SVMs) do not easily apply to the highly imbalanced whole-genome predictive setting. Reimplementing, adapting, and optimizing the previously described models to an entirely new domain (i.e., preexisting chromatin) was not achievable in the short timespan afforded to this revision.

Nevertheless, the reviewer's point is still valid; if the previously described methods could be adapted to the setting described in our manuscript, would Bichrom offer any advantages or additional insights? Bichrom performs two related tasks: **1)** Inferring the relative contributions of DNA sequence and preexisting chromatin features toward explaining a TF's global binding landscape (Manuscript Figures 2 & 7); and **2)** Inferring the relative contribution of DNA sequence and preexisting chromatin features toward explaining a TF's binding at individual TF binding sites (Manuscript Figures 3, 4, 5, 6).

In relation to task 1 (relative contributions at the global level), we agree with the reviewer that previously described methods may be applicable to analyze the marginal predictive capacity of preexisting chromatin features. However, Bichrom offers several improvements over these methods, especially when considering preexisting chromatin (summarized below). We have now added additional discussion of these methods and Bichrom's advantages in our manuscript's Discussion.

1. Tsai, *et al.* (PMID: 26291518) estimate the contributions of sequence motifs, concurrent chromatin state, and DNA shape to TF occupancy by calculating whether the distributions of these features are significantly different at TF bound vs. TF unbound sites. However, they do this in the context of the relatively small yeast genome. It's not clear how such a strategy would translate to larger mammalian genomes, where most features would look depleted in the larger negative sets.
2. Kumar, *et al.* (PMID: 26818008) infer the capacity of concurrent chromatin to predict TF binding by training a model to discriminate between the top 20% and bottom 20% of TF ChIP-seq binding peaks. This method assumes that if chromatin accessibility is correlated with TF binding, stronger ChIP-seq peaks are more likely to display higher accessibility. This assumption does not always hold when dealing with preexisting chromatin features. For instance, while the top and bottom 20% of ChIP-seq peaks for Ascl1 are clearly different in their concurrent chromatin accessibility, we observe very little difference in preexisting chromatin accessibility levels at these sites (Reviewer Figure 1). In contrast, Bichrom has a multi-step training strategy that uses chromatin-matched data to train a sequence model, while also enabling the chromatin model to access approximately 1 million unbound sequences uniformly sampled from the genome to capture the background chromatin distribution.
3. Arvey, *et al.* (PMID: 22955984) use a similar strategy as we did in assessing global contributions: they trained a model to predict TF binding sites with sequence and chromatin features, and

compared its performance to a model trained using sequence alone. However, their training and evaluation setting is quite different to ours. Their sequence model uses balanced sets of positive and negative examples, and their chromatin model uses a positive:negative ratio of 1:6. In contrast, our model is trained and evaluated genome-wide, enabling a holistic assessment of relative global contributions.

- We also note that Bichrom's CNN-LSTM architecture is significantly more expressive than the sequence models used in these other studies, and thus provides a stronger baseline estimate of sequence contributions toward global TF binding prediction. Tsai, *et al.* use only PWM scores, which are not sufficient to model the complexity of TF binding sites, especially in mammalian genomes. Kumar, *et al.* and Arvey, *et al.* use SVMs trained on more complex oligonucleotide representations, but such approaches have been previously demonstrated to be outperformed by CNN-based neural networks on genome-wide TF binding prediction tasks (e.g. Alipanahi, *et al.* PMID: 26213851).

Reviewer Figure 1: The distribution of **A)** preexisting chromatin accessibility and **B)** concurrent chromatin accessibility at the top 20% versus bottom 20% of Ascl1 binding peaks.

In relation to task 2 (relative contributions at individual sites), our work is completely novel to the best of our knowledge. Bichrom acts as a generalized additive model (GAM), where non-linear transformations are applied to sequence and chromatin features at each input training window via neural networks, and the transformed inputs are additively combined to predict TF binding. This allows us to quantify the contributions of sequence and chromatin predictors at individual binding sites, without sacrificing model complexity. No previous methods have performed such an analysis (even for concurrent data).

Tsai, *et al.* use a Random Forest (RF) to train a joint sequence + concurrent chromatin model, whereas Kumar, *et al.* use both unified RFs and SVM models trained on sequence and chromatin features. Neither RFs nor SVMs are additive, and hence the contribution of each feature type (sequence or chromatin) to the predictive output is not directly interpretable at individual sites. In contrast, Bichrom is designed to be directly interpretable (for example: in Figure 3A, 4C, Supp. Fig. 11B we show that TFs bind stronger sequence signatures at previously inaccessible chromatin).

Arvey, *et al.* train completely independent SVMs for sequence and chromatin features and add the outputs. While this approach provides separable sequence and chromatin models that could be applied

to individual binding sites, the fact that they are trained independently makes interpretation of relative contributions challenging. For example, an independently trained sequence SVM could learn sequence features that are generally correlated with active chromatin regions. In contrast, Bichrom's training scheme actively discourages the sequence subnetwork from learning features that are redundant with information provided by preexisting chromatin features.

Finally, in terms of methodological novelty, it is worth noting that Bichrom's design fits with a recently formalized concept of a Neural Additive Model (<https://arxiv.org/abs/2004.13912>), which is a linear combination of neural networks that each attend to a single input feature. The Neural Additive Model concept was described after submission of our manuscript, but lends formal justification to our novel approach. We now discuss the merits of the cited methods in our manuscript, their potential applicability to preexisting data, caveats to consider when modeling preexisting data, and finally the novel contribution of our work. Specifically, we have clarified that the novelty of Bichrom lies in its design that enables per-site interpretability.

2. In the new presentation, the authors have focused on the role of existing histone environment in a newly expressed TF. They're using chromatin marks at a time T0 in a model to predict ChIP data of a TF at a later time T1. It does not, however, show the role (if any) of chromatin marks at time T1 in the occupancy of the TF at T1. For a pioneer TF like Ascl1, this is an important part of Ascl1's binding mechanism.

We respectfully disagree with the reviewer's viewpoint here. The chromatin environment at T0 is important for determining the binding of an induced TF at T1. In contrast, the chromatin environment at T1 is consequential; i.e., it follows as a result or effect of the TF's binding and regulatory activities. Assessing correlations between TF binding at T1 and chromatin marks at T1 (what we call "concurrent" chromatin features) is the focus of a large number of other studies, including the three mentioned by the reviewer in the previous comment. While analyzing the relationships between TF binding and concurrent chromatin features is extremely useful for predictive modeling and other applications, it cannot be used to assess the mechanisms driving TF binding specificity. For example, if we just see a correlation between a TF's binding sites and a concurrent histone modification, it's impossible to infer whether the histone modification is causing TF binding or whether TF binding is causing the histone modification. Therefore, we do not generally consider the correlations between concurrent chromatin features and TF binding in this study.

However, for the purposes of model performance comparison, we did train a version of Bichrom to predict Ascl1 binding using concurrent chromatin features (Supp. Figure 3B). We also note that we have carried out analyses of the relationships between Ascl1 binding and concurrent chromatin features in a previous study (Aydin, *et al.* 2019). Notably, many of the Ascl1 binding sites that are inaccessible in mEB cells (T0) become accessible and marked by active histone modifications after Ascl1 binding (T+12hrs).

3. The authors noted "Thus, our approach may offer a metric for quantifying the "pioneering" activity of a TF; those that display less dependence on preexisting chromatin may be more likely to override those landscapes." Did they mean pioneer TFs to display less dependence on preexisting chromatin or the opposite? Ascl1 is a pioneer TF, but their model showed a major dependence on preexisting chromatin.

We should expect pioneer TFs to display fewer dependencies on preexisting chromatin landscapes; they should be able to bind to both accessible and inaccessible sites. However, as noted by the reviewer, Bichrom finds that Ascl1 binding is predicted to some degree by preexisting chromatin features. We see

now how this observation may be confusing given Ascl1's known pioneering activity. We have included additional discussion addressing this observation.

Briefly, despite being a pioneer TF, Ascl1 binding displays some dependence on the preexisting chromatin landscape. Several recent studies of pioneer TFs such as FoxA, Oct4, and GATA in human and mouse cells have also shown that despite pioneering activity (i.e., binding at relatively pre-inaccessible chromatin) pioneer TFs can still depend on the preexisting chromatin landscape. See Cernilogar, *et al.* (PMID 31350899) for one recent example. In addition, pioneer TFs bind distinct sets of sites when induced in different cell types, demonstrating that the preexisting regulatory landscape must play a role in their binding specificity.

Importantly, Bichrom analysis suggest that Ascl1 is less dependent on preexisting chromatin than other analyzed TFs. For example, Brn2 and Ebf2 show higher degrees of dependence on preexisting accessibility. On the other hand, another known pioneer TF, Onecut2, displays almost no dependence on preexisting accessibility (Manuscript Figure 6A). One of the conclusions of our work is that dependencies on preexisting chromatin may vary across TFs and perhaps even across cellular contexts. In other words, TFs may display a range of pioneering activities as opposed to it being an all or nothing property. Bichrom offers a way to compare these relative pioneering abilities. Thus, the wording in the sentence was what we intended. However, we have adjusted that sentence to provide more clarity.

4. From Fig 7, are the models useful where auPRC were very bad?

We decided to include these models since the magnitude of the auPRC can be artifactually lower for TFs with very few ChIP-seq peaks (a small positive set) despite such models having true positive and false positive rates which are comparable to TFs with a larger number of peaks and higher auPRCs (see Methods). However, we agree with the reviewer's suggestion that models displaying low auPRC values should be interpreted with caution. We have added an appropriate caveat in the manuscript Discussion.

Minor comment:

The paper "Chromatin Landscape Dictates HSF Binding to Target DNA Elements" by Guertin, M. J. & Lis, J. T. appears twice in the bibliographic references. The authors need to check if this happened with any other cited manuscript.

We thank the reviewer for catching this erroneous citation. We have removed the additional citation and reviewed our citation list.

Reviewer #2

I suggest including a short descriptive title for each figure. Overall, I am satisfied with the revised manuscript. Good job.

We thank the reviewer for reviewing our manuscript and for their support. We have added a short descriptive title for each figure.